# Tumor microenvironment derived exosomes pleiotropically modulate cancer cell metabolism

Hongyun Zhao[1,2], Lifeng Yang[1,2], Joelle Baddour[1,2], Abhinav Achreja[1,2], Vincent Bernard[3], Tyler Moss[4], Juan C Marini[5], Thavisha Tudawe[2], Elena G Seviour[4], F Anthony San Lucas[3], Hector Alvarez[3], Sonal Gupta[3], Sourindra N Maiti[6], Laurence Cooper[6], Donna Peehl[7], Prahlad T Ram[4], Anirban Maitra[3], Deepak Nagrath[1,2,8]*

[1]Laboratory for Systems Biology of Human Diseases, Rice University, Houston, United States; [2]Department of Chemical and Biomolecular Engineering, Rice University, Houston, United States; [3]Departments of Pathology and Translational Molecular Pathology, Ahmad Center for Pancreatic Cancer Research, University of Texas MD Anderson Cancer Center, Houston, United States; [4]Department of Systems Biology, University of Texas, MD Anderson, Houston, United States; [5]Baylor College of Medicine, Houston, United States; [6]Department of Pediatrics, University of Texas MD Anderson Cancer Center, Houston, United States; [7]Department of Urology, School of Medicine, Stanford University, Stanford, United States; [8]Department of Bioengineering, Rice University, Houston, United States

*For correspondence: dn7@rice.edu

Competing interests: The authors declare that no competing interests exist.

**Abstract** Cancer-associated fibroblasts (CAFs) are a major cellular component of tumor microenvironment in most solid cancers. Altered cellular metabolism is a hallmark of cancer, and much of the published literature has focused on neoplastic cell-autonomous processes for these adaptations. We demonstrate that exosomes secreted by patient-derived CAFs can strikingly reprogram the metabolic machinery following their uptake by cancer cells. We find that CAF-derived exosomes (CDEs) inhibit mitochondrial oxidative phosphorylation, thereby increasing glycolysis and glutamine-dependent reductive carboxylation in cancer cells. Through 13C-labeled isotope labeling experiments we elucidate that exosomes supply amino acids to nutrient-deprived cancer cells in a mechanism similar to macropinocytosis, albeit without the previously described dependence on oncogenic-Kras signaling. Using intra-exosomal metabolomics, we provide compelling evidence that CDEs contain intact metabolites, including amino acids, lipids, and TCA-cycle intermediates that are avidly utilized by cancer cells for central carbon metabolism and promoting tumor growth under nutrient deprivation or nutrient stressed conditions.

## Introduction

The understanding of interaction mechanisms between cancer cells and the tumor microenvironment (TME) is crucial for developing therapies that can arrest tumor progression and metastasis. Recent studies have identified the TME as a key player in regulating cancer cell growth (*Whiteside, 2008*). Although the TME is comprised of a variety of cell types including cancer-associated fibroblasts cells (CAFs), immune cells, and angiogenic elements, CAFs are the major constituent of the TME in many cancers (*Whiteside, 2008*; *Allinen et al., 2004*; *Feig et al., 2012*). Accumulating evidence suggests that paracrine signals from cancer cells can both recruit and activate CAFs within the TME, and

**eLife digest** Cancer cells behave differently from healthy cells in many ways. Healthy cells rely on structures called mitochondria to provide them with energy via a process that requires oxygen. However cancer cells don't rely on this process, and instead release energy by breaking down sugars outside of the mitochondria. This may explain why cancer cells are able to thrive even when little oxygen is available.

Cancer cells also interact with neighboring cells called fibroblasts, which are a major part of a tumor's microenvironment, and recruit them into the tumors. The fibroblasts communicate with cancer cells, in part, by releasing chemical messengers packaged into tiny bubble-like structures called exosomes. Recent studies have suggested that these exosomes may help cancer cells to thrive, but there are many questions remaining about how they might do this.

Now, Zhao et al. show that the fibroblasts smuggle essential nutrients to cancer cells via the exosomes and disable oxygen-based energy production in cancer cells. First, exosomes released by cancer-associated fibroblasts from people with prostate cancer were collected and marked with a green dye. Next, the green-labeled exosomes were mixed with prostate cancer cells, and shown to be absorbed by the cells. Oxygen-based energy release was dramatically reduced in the exosome-absorbing cells, and sugar-based energy release increased.

Next, Zhao et al examined the contents of the exosomes, and found that they contain the building blocks of proteins, fats, and other important molecules. Next, the experiments revealed that both prostate cancer and pancreatic cancer cells deprived of nutrients can use these smuggled resources to continue to grow. Importantly, this process did not involve the protein Kras, which previous studies had show helps cancer cells absorb nutrients. These findings suggest that preventing exosomes from smuggling resources to starving cancer cells might be an effective strategy to treat cancers.

contribute to their activation (*Whiteside, 2008*; *Liao et al., 2009*; *Orimo et al., 2005*; *Chung et al., 2006*). Although CAFs have been associated with tumor growth, progression, and metastasis through intercellular communications with cancer cells; little is known about their role in inducing metabolic reprogramming in cancer cells.

Studies have shown that extracellular vesicles known as exosomes can facilitate crosstalk between cancer and stromal cells in the TME. Exosomes have emerged as a vital communication mechanism between different cell types in the TME. Exosomes carry information from one cell to another and reprogram the recipient cells (*Gangoda et al., 2015*), and recent findings report that exosomes harbor the potential to regulate proliferation, survival and immune effector status in recipient cells. Exosomes range between 30–100 nm in diameter, have a bilayered membrane (*Johnstone et al., 1987*) and express surface marker such as CD63 (*Christianson et al., 2013*). Recent studies indicate that they contain proteins, nucleic acids and miRNAs (*Ekström et al., 2012*; *Costa-Silva et al., 2015*; *Simons and Raposo, 2009*). Most of the current studies are focused on cancer cell secreted exosomes; and little is known about CAF-derived exosomes (CDEs) and their metabolic influence on cancer cells. Although it has been shown that CAFs can induce metabolic reprogramming in cancer cells (*Brauer et al., 2013*), the contribution of CDEs in this phenomenon, if any, has not been elucidated.

Here, we report a novel regulation of cancer cell metabolism in prostate and pancreatic cancers mediated by CDEs. Our results demonstrate that patient-derived CDEs reprogram cancer cell metabolism through disabling mitochondrial oxidative metabolism and providing *de novo* 'off the shelf' metabolites through exosomal cargo. Specifically, we find that inhibition of mitochondrial oxidative phosphorylation by CDEs is associated with a compensatory increase in glycolysis. Interestingly, the inhibition of electron transport chain by CDEs significantly increased glutamine's reductive carboxylation for biosynthesis in cancer cells. Further, we demonstrate through isotope tracing and intra-exosomal metabolomic experiments that exosomes act as a source of metabolite cargo carrying lactate, acetate, amino acids, TCA cycle intermediates, and lipids; and these metabolites are utilized by recipient cancer cells for proliferation, precursor metabolites and replenishing levels of TCA

cycle metabolites. Notably, we demonstrate in wild-type and activated Kras-expressing pancreatic cancer cells that the metabolite cargo delivery mechanism by exosomes is similar to macropinocytosis, albeit without the previously described dependence on oncogenic Kras signaling (*Commisso et al., 2013*). Our results reveal a novel metabolism-centric regulatory role of TME-secreted exosomes in cancers and we uncover the underlying mode of action of this regulation. These findings can lead to novel therapeutics targeting communication between cancer cells and their microenvironment.

## Results

### CDEs are internalized by prostate cancer cells

To illustrate that CAFs secrete exosomes, and that cancer cells internalize these exosomes, we first isolated exosomes from conditioned media obtained from patient-derived prostate CAFs. The particle size analysis of isolated exosomes showed particles with size distribution from 30 to 100 nm (*Figure 1A*), which is consistent with previous observations (*Xiao et al., 2014*). Since exosomes are below the size range to allow direct detection by flow cytometry, we confirmed exosomes' expression of CD63, a surface antigen marker, through flow analysis of Dynabeads conjugated with anti-CD63 antibody (*Figure 1B*). To examine if CDEs are taken up by prostate cancer cells (PC3), we pre-labeled CDEs with PKH green dye and added them to PC3 cells for 3h and analyzed their internalization by cancer cells. As indicated by shift in the peaks, CDEs are indeed taken up by cancer cells (*Figure 1C*). Examination by fluorescence microscopy also confirmed the uptake of PKH red labeled exosomes by PC3 cells, evidenced through colocalization of red fluorescence and DAPI (*Figure 1D*). Furthermore, we estimated the saturable concentration of CDEs taken up by cancer cells (*Figure 1E*). Hence, in subsequent experiments we used 200 µg/ml of CDEs as the working concentration (*Zhu et al., 2012*).

### CDEs downregulate mitochondrial function of prostate cancer cells

Since CAFs have been shown to regulate cancer cell growth (*Liao et al., 2009*), we first examined influence of CDEs on cancer cell proliferation. We isolated exosomes from the conditioned media of CAFs derived from a prostate cancer patient and cultured prostate cancer cells in the presence of freshly isolated exosomes. CDEs enhanced proliferation of PC3 cells with increasing exosomes concentration (*Figure 2A*). To determine whether CDEs induce metabolic rewiring in cancer cells, we cultured PC3 cells in CDEs for 24 hr and measured the oxygen consumption rate (OCR) with increasing amounts of exosomes. Surprisingly, we observed that basal oxidative phosphorylation (OXPHOS, indicated by OCR) was significantly inhibited with increasing concentration of CDEs added to PC3 cells (*Figure 2B*). To ascertain whether the inhibition of mitochondrial respiration of cancer cells is specific to CDEs and to prove similar behavior is not exhibited with exosomes derived from other cells, we isolated exosomes from prostate cancer cell line (PC3), human fibroblasts (IMR-90), and also used blank media for isolation method control (*Figure 2—figure supplement 1*). As seen in the figure, exosomes from control conditions were ineffective in modulating cancer cells' OCR. To expand our observations, we next isolated exosomes from three independent prostate cancer patient CAFs and cultured four prostate cancer cell lines (PC3, DU145, 22RV1 and E006AA) in presence and absence of the exosomes (*Figure 2C*). Remarkably, exogenous addition of CDEs reduced OCR in all prostate cancer cell lines. To confirm if this reduction of OCR in cancer cells was indeed because of uptake of exosomes, we added the endocytosis inhibitor Cytochalasin D (CytoD) in PC3 culture media along with CDEs. CytoD has been shown to inhibit exosome uptake in various cell systems (*Casella et al., 1981*; *Feng et al., 2010*). Notably, CytoD could partially rescue this reduction of OCR in PC3 cells, thus confirming the CAF exosomes mediated reduction of OCR in cancer cells (*Figure 2D*).

To conclusively associate CDEs induced metabolic reprogramming with metabolic content of exosomes, we verified if synthetic liposomes (DOPC/CHOL liposomes labeled with DiO, size 85-110 nm; DOPC: 1,2-dioleoyl-sn-glycero-3-phosphocholine, CHOL: cholesterol) with a size distribution similar to exosomes could similarly modulate cancer cells. Our data suggests that liposomes did not alter cell proliferation, OCR and ECAR in both prostate (PC3) and pancreatic cancer cells (MiaPaCa-2 and BxPC3) (*Figure 2—figure supplement 2*). These results implicate metabolic content of exosomes

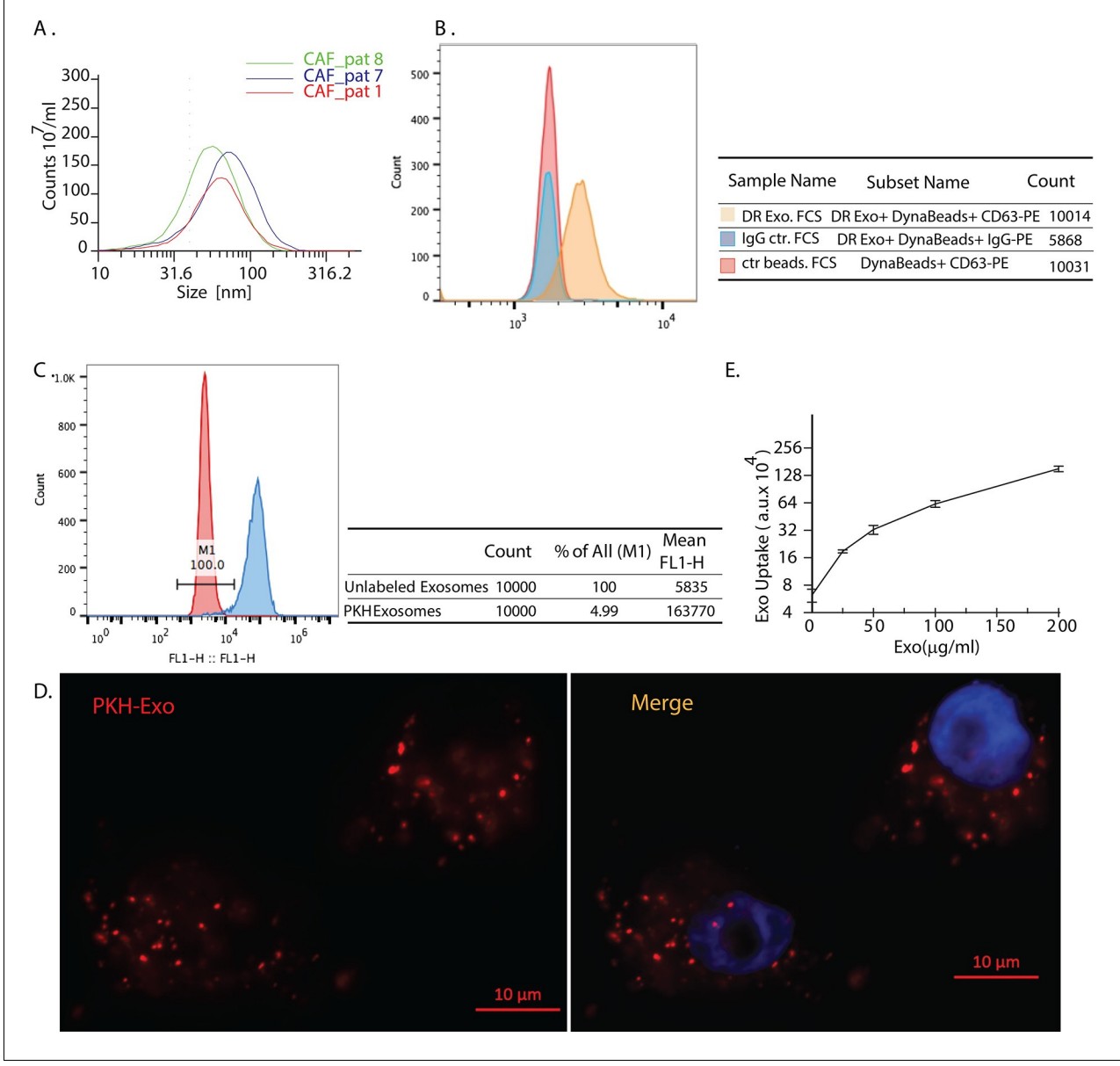

**Figure 1.** Exosomes secreted by CAF-derived from prostate cancer patients are internalized by prostate cancer cells. (A) Size analysis of stromal exosomes. Three samples of exosomes derived from prostate cancer patient CAFs were analyzed with the Zetaview instrument. The profiles indicate that the size distribution of exosomes is within the range of 30-100 nm. For exosomes isolation, conditioned medium was obtained from CAFs cultured with exosomes-depleted FBS. (B) Flow analysis of CAF exosomes bound to Dynabeads conjugated with anti-CD63 antibody (anti-CD63) or an irrelevant control antibody (anti-Rabbit IgG antibody, Rb IgG). The graph and table show that these microvesicles express CD63, an exosome surface antigen biomarker. (C) Flow cytometry analysis shows uptake of CAF exosomes by prostate cancer cells. Prostate cancer cells were incubated with PKH67-labeled stromal exosomes for 3 hr. Freshly prepared exosomes were used in this and subsequent experiments. Exosome-depleted serum was used for cell culture. (D) Representative fluorescence image shows CAFs exosomes were uptaken by prostate cancer cells. Prostate cancer cells were incubated with PKH26-labeled CAFs exosomes for 3 hr. Blue, cell nuclei; Red, PKH-Exo. (E) Flow cytometry analysis shows saturable uptake curve of CAFs secreted exosomes in prostate cancer cells. (n=4).

towards observed changes in CDEs-induced increase in cancer cell proliferation, mitochondrial dysfunction and increased glycolysis.

The mitochondrial respiratory capacity inhibition in prostate cancer cells by prostate CAF-exosomes was further confirmed by measuring maximal and reserve mitochondrial capacity using oligomycin, the protonophoric uncoupler FCCP, and the electron transport inhibitor rotenone. Both

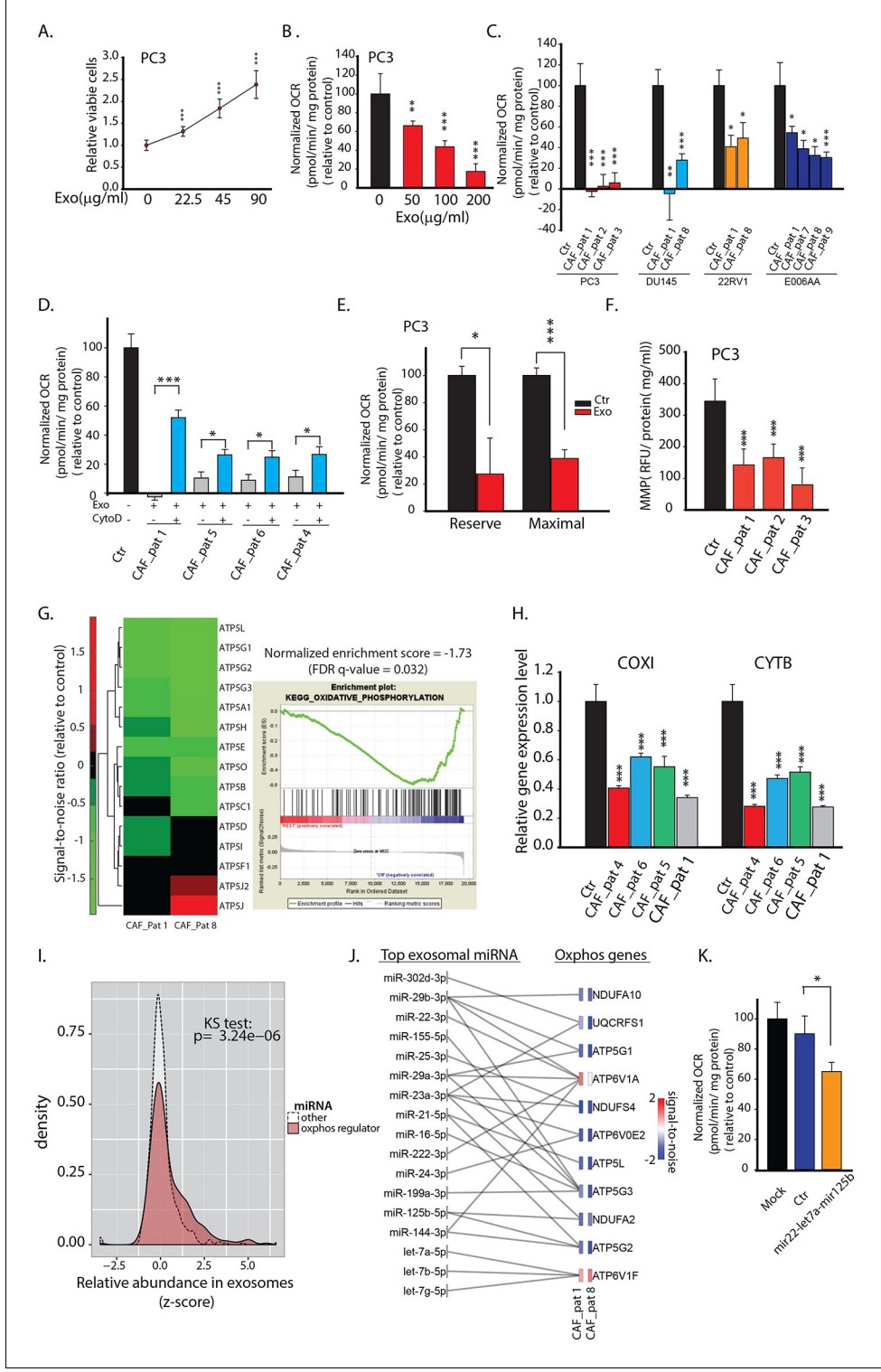

**Figure 2.** CDEs increase proliferation of prostate cancer cells but significantly downregulate their mitochondrial function. (**A**) Effect of CAFs-derived exosomes on viability of prostate cancer cells, 48h culture period (PC3) (n≥9). (**B**) Prostate cancer cells show reduced basal mitochondrial oxygen consumption rate (OCR) when cultured with range of concentrations of CDEs for 24 hr. Basal OCR is a measure of OXPHOS activity. The OCR was normalized with protein content inside cells. PC3 cells were cultured with patient-1 derived CAFs' exosomes (n≥9). (**C**) Basal OCR was measured for PC3, DU145, 22RV1, E006AA prostate cancer cell lines cultured with patient derived CDEs and control conditions. Six patient-derived CAFs were used for exosomes isolation. (n≥9). (**D**) OCR of prostate

*Figure 2 continued on next page*

*Figure 2 continued*

cancer cells were measured after 24 hr culture with and without CDEs. Cytochalasin D (CytoD), an inhibitor of exosomes uptake through actin depolymerization, rescues reduced OCR in prostate cancer cells when cultured with CAFs exosomes. CytoD disturbs actin filament inside cells, thus inhibit phagocytosis. CytoD concentration of 1.5 µg/ml was used. (n≥5). (E) Maximal and reserve mitochondrial capacities were measured using FCCP and antimycin. Maximal OCR is maximal capacity of mitochondrial OCR. (n≥9). (F) Role of CAFs secreted exosomes in regulating mitochondrial membrane potential (MMP) of prostate cancer cells. MMP is an important indicator of mitochondrial functions. (n≥5). (G) Reduced OXPHOS genes expression in cancer cells cultured with exogenous CDES. (H) qPCR results show that mitochondrial OXPHOS genes of prostate cancer cells were downregulated when cultured with CDEs. (n=3). (I) Most abundant miRNAs targeting OXPHOS genes were abundant in CAFs exosomes. (n=4). (J) miRNAs in CAFs exosomes targeting specific OXPHOS genes. Nanostring was used to measure miRNA expression levels in stromal exosomes. (n=4). (K) OCR of PC3 were measured after transfection of targeted miRNAs together into cells. (n=5). miRNAs were transfected into cells according to the manufacturer's protocol (Lipofectamine 2000 Transfection Reagent, Thermofisher). Cells were seeded in 6-well plate for 24 hr. Transfection was performed followed by incubation for 48 hr. Cells were then reseeded onto Seahorse plates for OCR measurements after the cells were attached. Data information: data in (A), (B), (C), (F), (H) are expressed as mean ± SD, data in (D), (E), (K) are expressed as mean ± SEM;*p<0.05, **p<0.01, ***p<0.001. *Figure 2—figure supplement 1–2*.

The following figure supplements are available for figure 2:

**Figure supplement 1.** Specificity of CDEs in regulating mitochondrial respiration of cancer cells is demonstrated.

**Figure supplement 2.** Effects of synthetic liposomes in regulating PC3 metabolism and viability.

maximal and reserve mitochondrial capacity of cancer cells were significantly reduced in presence of CDEs (*Figure 2E*). These results suggest that CAFs downregulate mitochondrial OXPHOS in cancer cells and CDEs play a key role in this reprogramming. To further examine the effect of these exosomes on mitochondrial activity, we measured the mitochondrial membrane potential of PC3 cells with and without exosomes. As seen in *Figure 2F*, exogenous addition of CDEs significantly reduced mitochondrial membrane potential within cancer cells.

To unravel the mechanism behind OCR reduction in cancer cells by CDEs; we performed microarray and q-PCR analysis to estimate the changes in mitochondrial gene expression levels of PC3 cells with and without exogenous CDEs (*Figures 2G,H*). Microarray data revealed that transcript levels for OXPHOS related ATP synthase complex genes were downregulated in cells cultured with CDEs. Gene Set Enrichment Analysis (GSEA) on microarray data corroborates these observations by estimating a negative enrichment score (p-value <0.05) for the entire set of 109 OXPHOS-related genes in cancer cells cultured in presence of CDEs relative to control (*Figure 2G*). Furthermore, we found that both cytochrome B (CYTB) and cytochrome C oxidase I (COXI), which are components of Complexes III and IV of the electron transport chain, respectively, have lower transcript expression levels when PC3 cells were cultured with exogenous CDEs (*Figure 2H*).

It is well established that exosomes contain noncoding RNAs (e.g. miRNAs) which can serve as a communication mechanism between stromal and cancer cells. We measured miRNA levels in the CDEs to determine whether miRNA underlay the molecular mechanism by which the exosomes exert the metabolic changes we observed. We extracted miRNAs from purified CDEs from three patients and measured the levels of a panel of 800 human miRNAs using NanoString technology. We grouped the miRNA by whether or not they target genes involved in oxidative phosphorylation (miRNA-mRNA interactions taken from starBase v2.0, starbase.sysu.edu.cn). The relative abundance of miRNAs that target oxidative phosphorylation genes is higher in exosomes than most other miRNA as seen in the density distribution plot (*Figure 2I*). The top 30 most abundant miRNAs across the exosomes sampled, target one or more OXPHOS genes, which is validated by measuring the decrease in mRNA levels of these genes after treatment with exosomes (*Figure 2J*). The miRNAs and their targets that we have identified are based on experimental miRNA abundance data from Nanostring assays followed by miRNA target prediction integrated with AGO-CLIP-SEQ data (*Li et al., 2014*). In order to confirm the inhibition of OXPHOS through miRNA we co-transfected mir-22, let7a and mir-125b present in the group of miRNA targeting OXPHOS, in PC3 cells, and

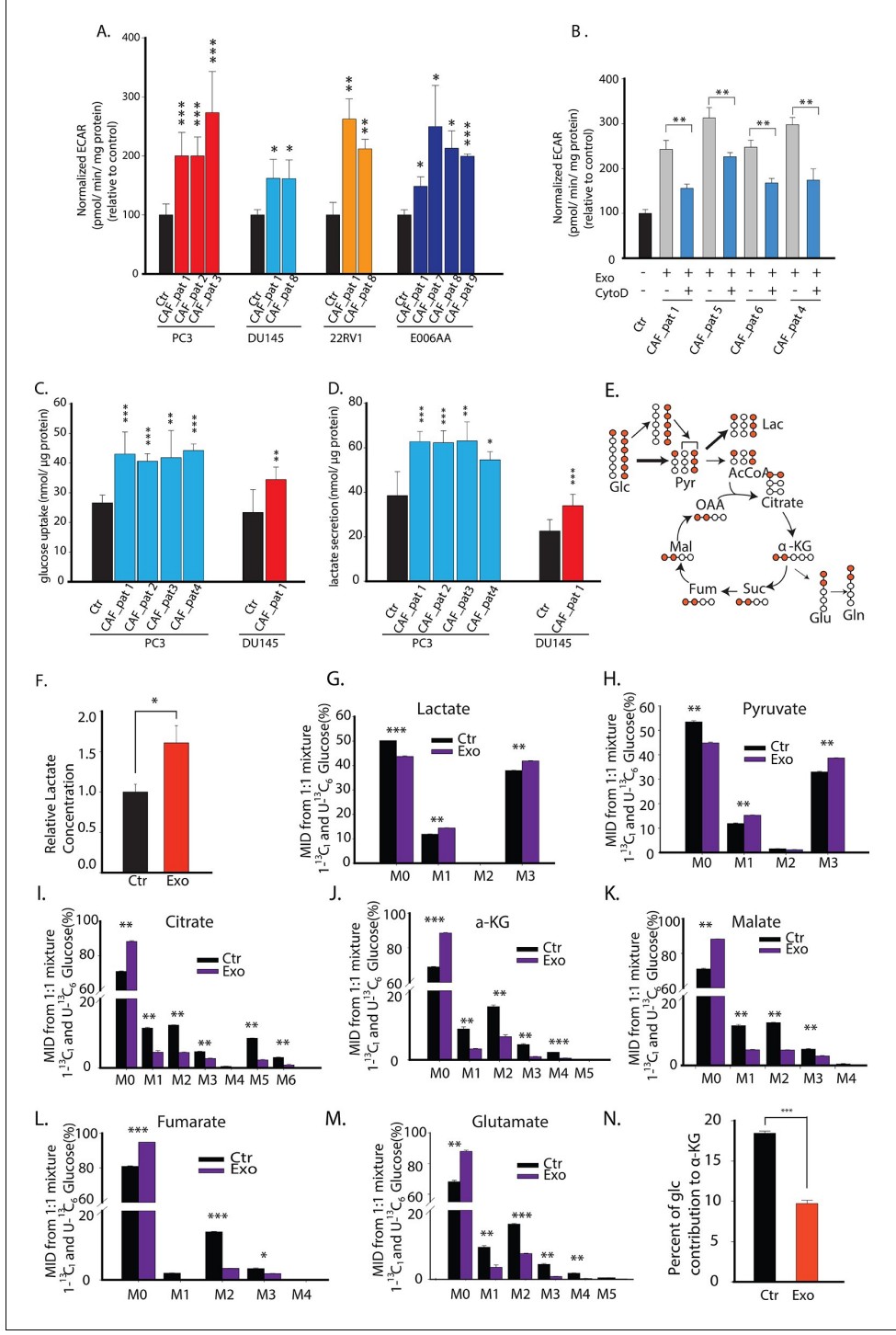

**Figure 3.** CDEs upregulate glycolysis in cancer cells. (**A**) Extracellular acidification rates (ECAR) of prostate cancer cells were measured after 24 hr culture with and without CAFs exosomes. ECAR is a measure of glycolytic capacity of cells. The ECAR was normalized with protein content inside cells. Four prostate cancer cell lines: PC3, DU145, 22RV1, E006AA were used. Six patients derived CAFs were used for exosomes isolation (n≥9). (**B**) ECAR of prostate cancer cells was measured. CytoD increased ECAR in prostate cancer cells when cultured with CAFs exosomes. CytoD concentration of 1.5 μg/ml was used. (n≥6). (**C,D**) Effect of CAFs-secreted exosomes on glucose uptake (**C**) and lactate secretion fluxes (**D**) in prostate cancer cells. (n=9). (**E**) Schematic of carbon atom transitions using 1:1 mixture of $^{13}C_6$ glucose and 1-$^{13}C_1$-labeled glucose. (**F**) Relative lactate abundances were measured using GC-MS in PC3 cells cultured with and without CAFs-secreted exosomes for 24 hr. (n=4). (**G–M**) Contribution

*Figure 3 continued on next page*

*Figure 3 continued*

of glucose towards TCA cycle metabolites and glycolysis is measured using the labeled glucose. Comparison of mass isotopologue distributions (MID) of lactate, pyruvate, citrate, α-ketoglutarate, malate, fumarate, and glutamate in PC3 cancer cells cultured with and without CAFs-secreted exosomes. (n=4). (**N**) Percentage of glucose contribution to α-ketoglutarate in PC3 cells with and without CAFs-secreted exosomes. (n=4). Data information: data in (**A,C** and **D**) are expressed as mean ± SD, data in (**B,F–N**) are expressed as mean ± SEM; *p<0.05, **p<0.01, ***p<0.001. *Figure 3—figure supplement 1*.

The following figure supplement is available for figure 3:

**Figure supplement 1.** Total ion currents of metabolites in PC3 with or without coculture of CDEs.

measured OCR (*Figure 2K*). In line with our hypothesis, we see a decrease in OCR in PC3 cells co-transfected with miRNAs. Although, the reduction of OCR is moderate, this is due to the technical limitation of co-transfection experiments which can only allow using a small subset of the miRNAs that target OXPHOS in PC3. In summary, these results suggest that CDEs reduced mitochondrial oxidative phosphorylation and induced metabolic alterations in cancer cells mimicking hypoxia-induced alterations.

## CDEs upregulate glucose metabolism in cancer cells

The above experiments showed that CDEs downregulate mitochondrial activity. We further investigated whether this reduced mitochondrial activity leads to increased glycolysis in cancer cells in presence of CDEs. We first measured levels of basal glycolysis (indicated by extracellular acidification rate, ECAR) in four prostate cancer cell lines in presence of exosomes from three independent prostate cancer patient CAFs (*Figure 3A*). As seen in the figure, these exosomes significantly increased glycolysis in cancer cells when compared to cancer cells cultured without exosomes. Notably, CytoD partially inhibited this increase of ECAR, thus confirming the role of exosomes in increase of glycolysis in cancer cells (*Figure 3B*). To expand our findings on the exosomes mediated increase of glycolysis in cancer cells; we measured both glucose uptake and lactate secretion in cancer cells cultured with and without exosomes for 24 h (*Figures 3C,D*). Consistent with above results, CDEs increased glucose uptake and lactate secretion when compared to cancer cells cultured without exogenously added exosomes.

To understand the underlying changes in metabolite abundances induced by CAF exosomes in cancer cells, we performed $^{13}$C GC-MS based isotope tracer analysis using a 1:1 mixture of U-$^{13}$C$_6$ glucose and 1-$^{13}$C$_1$ glucose (*Figure 3E–N*). GC-MS results are reported as mass isotopologue distributions (MIDs), which represent the relative abundance of different mass isotopologues of each metabolite; where M0 refers to the isotopologue with all $^{12}$C atoms and M1 and higher refer to heavier isotopologues with one or more $^{13}$C atoms derived from the tracer. It is well established that isotope tracer analysis can reveal the alterations in contributions of a substrate within a particular metabolic pathway (*Figure 3E*). We found that the CDEs increased the lactate levels in the cancer cells (*Figure 3F*, *Figure 3—figure supplement 1*). Further, the percentage of M3 pyruvate and M3 lactate was increased due to CDEs; however, there is a corresponding decrease in the percentage of M2 citrate and M3 citrate (*Figure 3G-I*). The increase of M3 pyruvate and M3 lactate indicates higher contribution of glucose to pyruvate and lactate in prostate cancer cells conditioned with CDEs. Moreover, there was a decrease of M0 pyruvate and M0 lactate with a corresponding increase of M1 pyruvate and M1 lactate, thus suggesting that the exosomes enhance glycolysis. The latter conclusion was based on the principle that M1 pyruvate is only produced by glucose-6-phosphate metabolized by phospho glucoisomerase. Consistent with the decreased OXPHOS observed in cancer cells due to CDEs, the percentage of M2 citrate, M2 α-ketoglutarate, M2 fumarate, M2 malate, and M2 glutamate was also significantly reduced in cancer cells in presence of CDEs (*Figure 3I–M*). In line with our observations, the percentage of $^{13}$C α-ketoglutarate from $^{13}$C labeled glucose was significantly reduced (*Figure 3N*). This further confirms that CDEs decreased the percentage contribution of glucose to a-ketoglutarate in cancer cells and instead diverted it towards lactate. The above results conclusively show that CDEs induce a Warburg type phenotype in cancer cells, by disabling normal oxidative mitochondrial function with a compensatory increase in glycolysis.

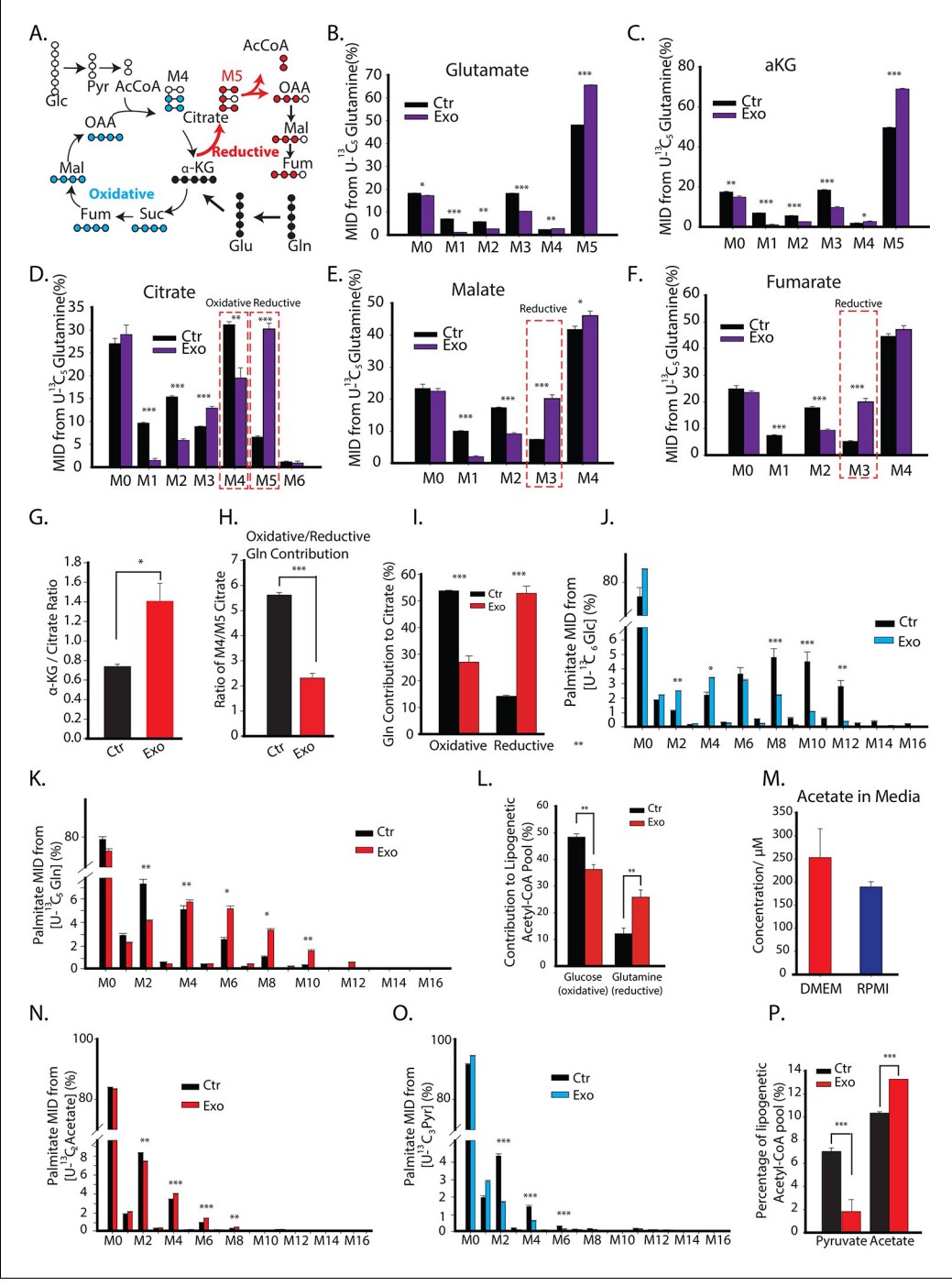

**Figure 4.** CDEs increase glutamine driven reductive carboxylation and lipogenesis in prostate cancer cells. (**A**) Schematic of carbon atom transitions using $^{13}C_5$ glutamine. Black color represents labeled carbon of glutamine before entering into TCA cycle. Blue color represents glutamine's direct effect on canonical TCA cycle and red color represents glutamine's effect on TCA cycle through reductive carboxylation. (**B–F**) Mass isotopologue distribution (MID) of glutamate, α-ketoglutarate, citrate, malate, and fumarate in PC3 cancer cells cultured with and without CDEs in U-$^{13}C_5$ glutamine (n=4). (**G**) Ratio of α-ketoglutarate and citrate pools in PC3 cancer cells cultured with and without CDEs measured using GC-MS. Higher ratio correlates with higher glutamine driven reductive carboxylation (n=4). (**H–I**) Ratio of glutamine contribution to citrate via oxidative and reductive pathways. Lower ratio indicates higher reductive carboxylation. CDEs increased reductive glutamine metabolism in PC3 cells (**I**). Oxidative contribution to citrate is determined by calculating M4 citrate percentage; reductive contribution to citrate is determined by M5 citrate percentage (n=4). (**J**) Glucose contribution to palmitate synthesis in PC3 cells

*Figure 4 continued on next page*

*Figure 4 continued*

cultured with or without CAFs exosomes for 72 hr was measured using GC-MS (n=6). (**K**) Glutamine contribution to palmitate synthesis in prostate cancer cells with or without CAFs exosomes measured using GC-MS (n=6). (**L**) Isotopologue spectral analysis (ISA) of both glucose and glutamine contribution to lipid synthesis in PC3 cells under control or CAFs exosomes culture conditions. CAFs exosomes enhance reductive carboxylation to lipid synthesis. However, total percentage of glucose and glutamine contribution to palmitate is about 60%. (**M**) Acetate concentration in cancer cell culture medium. (**N**) Acetate contribution to palmitate synthesis in PC3 cells with or without CAFs exosomes. Acetate spiked concentration was 500 μM (n=4). (**O**) Pyruvate contribution to palmitate synthesis in PC3 cells with or without CAFs exosomes (n=4). (**P**) ISA analysis of both pyruvate and acetate contribution to lipid synthesis in PC3 cancer cells under control or CAFs exosomes culture conditions. CAFs exosomes enhance acetate contribution to palmitate synthesis. Data information: data in (**B–P**) are expressed as mean ± SEM,*p<0.05, **p<0.01, ***p<0.001.

The following figure supplement is available for figure 4:

**Figure supplement 1.** Schematic of isotopologue spectral analysis (ISA) method.

## CDEs enhance reductive pathway of glutamine metabolism in cancer cells

Glutamine serves as an anaplerosis substrate to fuel the TCA cycle for energy generation and also provides nitrogen for protein synthesis (*Dang, 2010*; *Wise and Thompson, 2010*; *DeBerardinis and Cheng, 2010*; *Daye and Wellen, 2012*; *Gaglio et al., 2009*; *Johnson et al., 2003*; *Rajagopalan and DeBerardinis, 2011*; *Shanware et al., 2011*; *Weinberg et al., 2010*). To further unravel the mechanistic links between disabled normal oxidative mitochondrial function in cancer cells by CDEs and its influence on cancer cells' mitochondrial metabolism, we analyzed glutamine's contribution to the TCA cycle metabolite pools in cancer cells using labeled U-$^{13}C_5$ glutamine (*Figure 4A*). Proliferating cells under both normoxia and hypoxia can utilize glutamine by oxidative metabolism and produce pyruvate through malic enzyme and further combine oxaloacetate with acetyl-CoA to form M4 citrate (obtained by condensing of labeled oxaloacetate obtained from glutamine and unlabeled acetyl CoA). Alternatively, proliferating cells under hypoxia have been reported to predominantly reductively carboxylate glutamine generated α-ketoglutarate through IDH 1/2 to generate M5 citrate (*Figure 4A*) (*Metallo, 2012*). M5 citrate is further catalyzed to M3 fumarate and M3 malate in this reductive glutamine metabolism. Our MID data reveals that addition of CDEs increased M5 glutamate and M5 α-ketoglutarate in cancer cells thereby indicating that exosomes enhance glutamine's entry into TCA cycle (*Figure 4B,C*). Notably, there was significant increase in M5 citrate, M3 fumarate and M3 malate in cancer cells in the presence of exogenously added exosomes thus suggesting that cancer cells rely critically on reductive glutamine metabolism when normal mitochondrial function is disrupted by stromal microenvironment (*Figure 4D–F*).

To obtain mechanistic understanding of CDEs induced increased reductive carboxylation in cancer cells; we measured the ratio of α-ketoglutarate to citrate abundance in cancer cells and found that exosomes increased this ratio significantly (*Figure 4G*). The increased ratio of α-ketoglutarate to citrate was recently shown to promote reductive glutamine metabolism and it correlated with reductive glutamine's contribution to citrate (*Fendt et al., 2013*). The inhibition of respiratory chain components, or hypoxic conditions, was found to increase this ratio. Consistent with previous reports, the ratio of M4/M5 citrate, which represents the ratio of glutamine to citrate through oxidative metabolism over reductive metabolism, confirmed our above results that there is a significant increase in glutamine's reductive metabolism in presence of exosomes (*Figure 4H–I*). This is further substantiated through significant increase in the percentage contribution of glutamine through reductive pathway in the TCA cycle in cancer cells in presence of exosomes when compared to control condition. Also, there was concomitant decrease in the percentage contribution of glutamine through oxidative pathway in the TCA cycle.

One of the key metabolic requirements of rapidly dividing cancer cells pertains to availability of adequate pool of fatty acids for enabling membrane synthesis. Therefore, to further investigate the effect of exogenous CDEs on nutritional substrates' incorporation into lipogenesis; we used U-$^{13}C_6$ glucose or U-$^{13}C_5$ glutamine to estimate their conversion to cytosolic acetyl-CoA, which is the

precursor for palmitate (fatty acid) synthesis. Similar to our earlier observation that CDEs reduced glucose contribution to TCA cycle metabolites in cancer cells, we found that exosomes also decreased glucose contribution to palmitate. This is evident from the shift in high mass isotopologues of palmitate to lower mass isotopologues derived from U-$^{13}$C$_6$ glucose, when cells are cultured with exosomes (*Figure 4J*). Conversely, there is a shift in the reverse direction, i.e. from low to high mass palmitate isotopologues derived from U-$^{13}$C$_5$ glutamine, in presence of CDEs (*Figure 4K*). To quantify the percentage contribution of these substrates to the lipogenic acetyl-CoA pool, we performed isotopologue spectral analysis (ISA) (*Metallo, 2012*; *Kamphorst et al., 2014*). In line with above results, ISA (*Figure 4—figure supplement 1*) indicated a significant decrease in the fraction of glucose contribution to lipogenic acetyl-CoA in cancer cells cultured with exosomes (*Figure 4L*). More importantly, there is a two-fold increase in glutamine's contribution to lipogenic acetyl-CoA via the reductive carboxylation pathway. Additionally, these intriguing results suggest that there are likely to be other sources apart from glucose and glutamine that contribute to fatty acid synthesis.

Recently it was shown that acetate could be an important source for lipogenic acetyl-CoA in cancer cells, especially under hypoxic conditions(*Kamphorst et al., 2014*). To ascertain if acetate contributed to fatty acid synthesis, we measured acetate content in media (*Figure 4M*). Indeed, acetate concentrations were detected in our cancer cell media at significant levels. Since pyruvate was available in significant amounts in our cancer cell culture media, we included it in our estimates for palmitate synthesis. In order to quantify contribution from these alternative sources, we performed tracer experiments with U-$^{13}$C$_3$ pyruvate and U-$^{13}$C$_2$ acetate in cancer cells cultured with and without exosomes. We found that their contribution to lipogenic acetyl-CoA is significantly lower when compared to that of glucose and glutamine. This is evident from the low mass isotopologues of palmitate generated when cells are cultured with U-$^{13}$C$_3$ pyruvate or U-$^{13}$C$_2$ acetate (*Figure 4N–O*) in both control and exosomes-treated conditions. Interestingly, we noticed from the shift in palmitate mass isotopologues that CDEs increased acetate contribution to lipids (*Figure 4N*) but decreased the pyruvate contribution (*Figure 4O*). ISA results confirm these observations, where cancer cells cultured with CDEs showed an increase in acetate's contribution with a concomitant decrease in pyruvate's contribution to palmitate production (*Figure 4P*). These experiments collectively substantiate that exosomes have a significant effect on fatty acid synthesis in cancer cells by switching the carbon source from the oxidative glucose pathway to glutamine via the reductive carboxylation pathway in the TCA cycle.

## Intra-exosomal metabolomics reveal that CDEs contain an 'off-the-shelf' pool of metabolite cargo

Exosomes are known to carry a complex cargo that includes proteins, lipids, and miRNAs (*Costa-Silva et al., 2015*; *Simons and Raposo, 2009*). Results from the previous sections indicate that exosomes may act as a source of metabolites and proliferating cancer cells use these metabolites for lipogenesis. To ascertain whether exosomes contain significant amount of *de novo* metabolites, we first measured lactate and acetate contained inside the prostate and pancreatic CDEs. We included intra-exosomal metabolic measurements of pancreatic CAFs in order to generalize our conclusions. Notably, we found high amounts of lactate and acetate in both prostate and pancreatic CDEs (*Figure 5A,B*). This suggests that exosomes can not only replenish TCA cycle metabolites but also act as source of lipids. Further, to prove these hypotheses we performed GC-MS analysis for intra-exosomal metabolites and found high concentrations of citrate and pyruvate along with significant presence of α-ketoglutarate, fumarate and malate (*Figure 5C*). To further expand our findings, we performed ultra-high-performance liquid chromatography (UPLC), and found markedly high levels of glutamine, arginine, glutamate, proline, alanine, threonine, serine, asparagine, valine, and leucine in prostate CAF exosomes (*Figure 5D*). Additionally, in pancreatic CDEs, we found high levels of glutamine, threonine, phenylalanine, valine, isoleucine, glycine, arginine, and serine (*Figure 5E*). Remarkably, through GC-MS analysis of intra-exosomal lipids, we found intact stearate (*Figure 5F*) and palmitate (*Figure 5G*) at high levels in both prostate and pancreatic CDEs. Our results offer definitive proof for the first time that exosomes harbor an 'off-the-shelf' pool of metabolite cargo, TCA cycle metabolites, amino acids, and lipids, which can fuel the metabolic activity of the recipient cells.

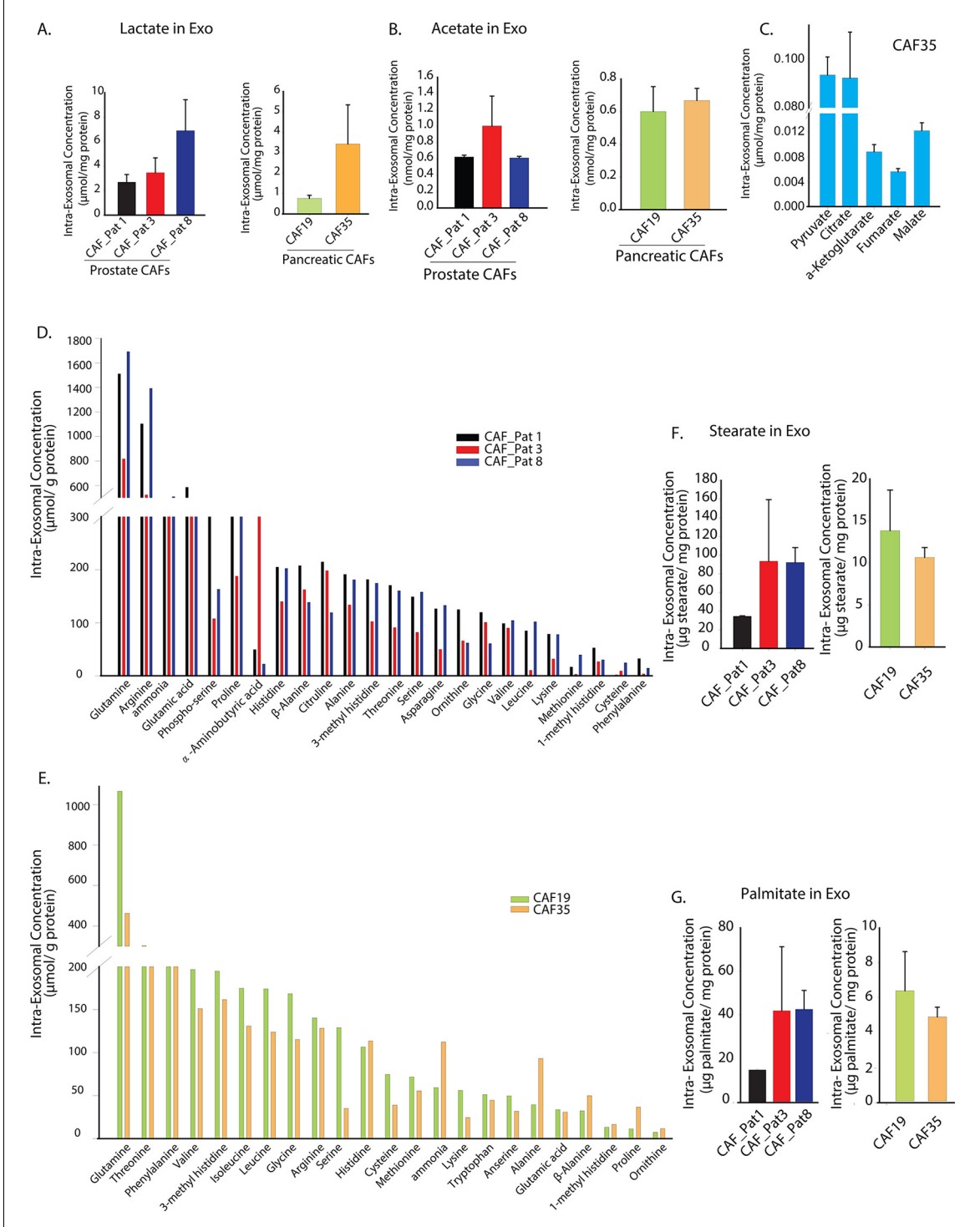

**Figure 5.** Prostate and pancreatic CAFs secreted exosomes carry metabolite cargo. Intra-exosomal lactate (**A**) and acetate (**B**) concentrations were measured in exosomes isolated from three prostate and two pancreatic CAFs using enzymatic assays. Intra-exosomal metabolites were extracted by methanol/chloroform method and protein concentration was used for normalization. (n=3). (**C**) TCA cycle metabolites, including pyruvate, citrate, α-ketoglutarate, fumarate and malate were measured using GC-MS in exosomes isolated from pancreatic CAF35. (n=3). (**D, E**) Amino acids were measured using ultra-high performance liquid chromatography (UPLC) inside CDEs (prostate CAFs: [**D**]; pancreatic CAFs: [**E**]). Significant levels of amino acids were detected inside CDEs. (n=3). (**F–G**) Stearate and palmitate were detected at high levels using GC-MS inside pancreatic and prostate CDEs. (*n* ≥ 3). Data information: data in (**A–C**), (**F–G**) are expressed as mean ± SEM.

## CDEs can supply amino acids to cancer cells in a manner similar to macropinocytosis

Recent studies have shown that macropinocytosis of circulating proteins (especially albumin) could supply amino acids to nutrient-deprived cancer cells (*Commisso et al., 2013*). Having established that CDEs could act as a source of metabolites, we further postulated that CDEs could act as source of TCA cycle metabolites for cancer cells. To establish whether metabolites contained in exosomes could fuel TCA cycle, we cultured patient-derived fibroblasts with $^{13}$C-labeled glucose, glutamine, pyruvate, leucine, lysine, and phenylalanine for 72 hr. We selected leucine, lysine and phenylalanine for labeling, because these were the most abundant amino acids in human serum albumin (*Saifer and Palo, 1969*). The extended timescale of 72 hr was adopted to allow detectable incorporation of labeling in proteins, amino acids, and lipids, and their compartmentalization within exosomal cargo. We then isolated labeled exosomes from the CAF culture spent medium. We postulated that nutrient deprived conditions will enhance cancer cells' dependence on the nutrient cargo in exosomes and therefore tested our hypothesis under both nutrient replete and nutrient deprived conditions. The isolated CDEs were then spiked into complete or nutrient-deprived (without lysine, leucine, phenylalanine, glutamine, and pyruvate) cultures of prostate cancer cells for 48h. The intracellular metabolites were isolated from cancer cells and their 13C enrichment was determined. Notably, our results substantiate that exosomes can supply metabolites to cancer cells under both complete and nutrient deprivation conditions (*Figure 6A*). However, and in concordance with hypothesis, we found that there is a significant increase in the contribution of CDEs to cancer cells' metabolites pools in nutrient deprived conditions, as compared to complete medium cultures. To definitively prove the direct export of metabolites by exosomes, we measured MIDs of metabolites in cancer cells when cultured with $^{13}$C labeled exosomes. We detected substantial labeling of intracellular amino acids in cancer cells, which included M5 glutamine, M6 lysine, and M6 leucine. We also detected M5 glutamate derived from mitochondrial glutaminolysis and labeled TCA cycle metabolites from labeled $^{13}$C-glutamine supplied by CDEs (*Figure 6B*). Further, it is important to note that substrates within CDEs will not be fully $^{13}$C labeled since 72 hr are not sufficient for CAFs to undergo at least one replication, and hence labeled metabolites in CDEs may be diluted with pre-existent unlabeled (M0) isoforms. This results in small, but significant levels of labeled metabolites within the cancer cells. Importantly, it provides a compelling proof-of-concept that CDEs can supply TCA cycle metabolites to cancer cells.

To estimate the contribution of CDEs in labeling cancer cells' metabolites, we determined MIDs of metabolites derived from isolated labeled CDEs and also from cancer cells cultured with labeled exosomes under nutrient deprivation conditions (*Figure 6—figure supplement 1A*). The deprivation and labeling conditions used were similar to *Figure 6A and B*. In order to quantify the contribution of metabolites from exosomes to cancer cells, we calculated the mean enrichment of $^{13}$C labeled amino acids and normalized them with their corresponding enrichment in exosomes. We observed that exosomes account for approximately 16% of phenylalanine, 14% of glutamine, 12% of lysine and 5% of leucine pools in PC3 cells (*Figure 6—figure supplement 1A*). Since MIDs are measured 48 hr after introduction of exosomes to cancer cells, several of the supplied metabolites would have been catabolized into other intermediates or incorporated into biomass precursors. Therefore, it is important to note that the contributions of essential amino acids estimated would be lower than their total contribution from exosomes over the course of 48 hr. In order to achieve higher detection of labeled isoforms, CAFs will have to be cultured with labeled substrates for multiple passages to completely replace unlabeled metabolite pools with their labeled isoforms. Nevertheless, these results substantiate that CDEs are key player in enriching metabolite pools in cancer cells under nutrient deprived conditions seen in the TME.

To evaluate the requirement of metabolites derived from exosomes for promoting tumor growth under nutrient deprivation conditions (without leucine, lysine, glutamine, pyruvate, and phenylalanine), we cultured cancer cells with exosomes under nutrient deprivation and complete media conditions with and without CytoD and heparin. Similar to CytoD, heparin has been recently shown to inhibit uptake of exosomes by cells (*Christianson et al., 2013*). CDEs were able to rescue reduction of proliferation under nutrient deprivation conditions (*Figure 6C*). However, this rescue effect is reduced to varying extents by adding CytoD, heparin and lysosomal degradation inhibitor chloroquine (*Figure 6C*). Similarly, addition of macropinocytosis inhibitor EIPA also counters the rescue of

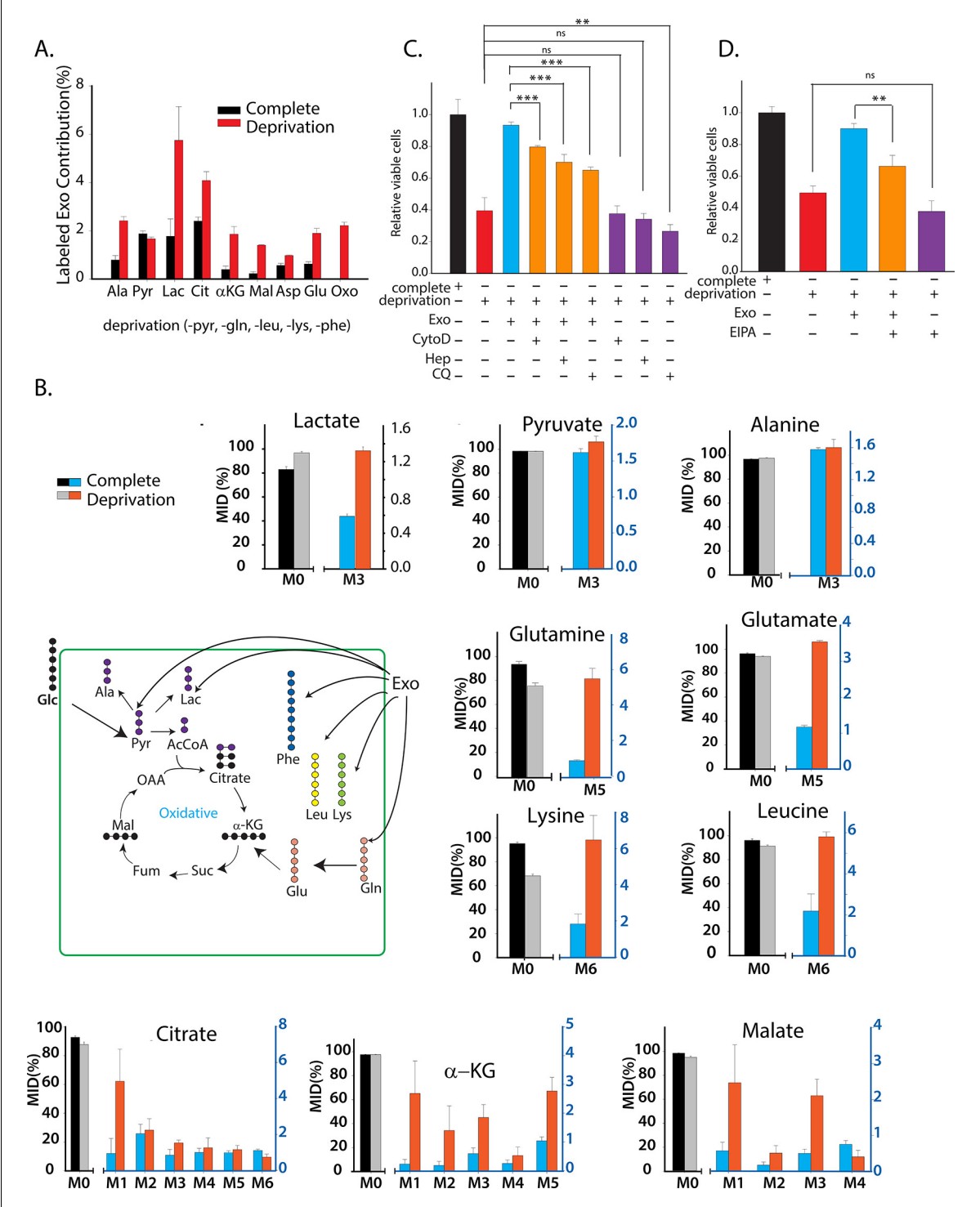

**Figure 6.** CDEs supply metabolites to cancer cells. To label metabolites, proteins and lipids in CAFs-secreted exosomes, CAFs were cultured in RPMI with labeled $^{13}C_3$ pyruvate (pyr), $^{13}C_5$ glutamine (gln), $^{13}C_6$ leucine (leu), 13C$_6$ lysine (lys), 13C$_9$-phenylalanine (phe) and U-$^{13}C_6$ glucose. After 72h of CAFs cultures, sufficient labeling was observed in metabolites, proteins and lipids contained in exosomes. Supply of metabolites to prostate cancer cells from labeled CDEs were measured under complete or deprivation medium cultures in culture media without labeling. (**A**) Percentage labeling (mean enrichment) observed in metabolites inside PC3 cells cultured with labeled CDEs. (n=4). Mean enrichment is calculated as

$ME = (\sum_{i=1}^{N} i \times M_i)/N.$ where $N$ is number of carbons in the metabolite and $M_i$ is abundance of (M+i) isotopologue (**B**) Mass fraction of heaviest labeled isotopologues of TCA cycle metabolites enriched by labeled CDEs, in prostate cancer cells cultured under complete or nutrient-deprived

*Figure 6 continued on next page*

Figure 6 continued

(without lys, phe, gln, pyr, leu) unlabeled medium (n=4). (C) Effect of CDEs on PC3 cell viability under deprivation (without lys, phe, gln, pyr, leu) conditions and exosome uptake inhibitors. CDEs rescue loss of PC3 cell proliferation under deprivation medium. CytoD, (1.5 µg/ml), heparin(50 µg/ml), and CQ (chloroquine, 20 µM) inhibited this rescue of viability under deprivation conditions n=10. (D) EIPA(25 µM) inhibited rescue of PC3 viability by CDEs under deprivation conditions, (n≥7). Data information: data are expressed as mean ± SEM,*p<0.05,**p<0.01, ***p<0.001.

The following figure supplement is available for figure 6:

**Figure supplement 1.** Contribution of metabolites from labeled exosomes to PC3 cells.

CDEs under deprivation (*Figure 6D*). These data suggest that uptake of exosomes and release of their cargo is necessary to rescue cell proliferation under nutrient deprived conditions. Taken together, these results provide evidence that CDEs can reprogram cancer cells' metabolism by acting as source of amino acids under nutrient depleted conditions in the TME.

## CDEs supply metabolites to pancreatic cancer cells via Kras-independent mechanism

We have shown that prostate CDEs can supply metabolites to prostate cancer cells. Our results were similar to the process of macropinocytosis, which was revealed as a mechanism to supply amino acids through extracellular proteins in oncogenic Ras-expressing pancreatic cancer cells. To expand the scope of our findings and understand whether Ras can similarly promote the supply of metabolites by CDEs in pancreatic cancer cells, we isolated exosomes from pancreatic CAFs cell line (CAF-19) and used them to study their metabolic influence in two pancreatic cancer cell lines: BxPC3 (wild type Kras) and MiaPaCa-2 (homozygous Kras). Since it was observed that CDEs supply metabolites to prostate cancer cells, we cultured both BxPC3 and MiaPaCa-2 cell lines, with and without CDEs under complete media and nutrient deprivation (without glutamine, leucine, lysine, phenylalanine, and pyruvate) conditions (*Figure 7A*). Notably, CDEs could rescue loss of proliferation in both cancer cell lines, thereby suggesting that internalization or uptake and supply of exosomes derived metabolites in cancer cells is Kras independent. Furthermore, this rescue of proliferation by CDEs in pancreatic cancer cell lines was inhibited by receptor mediated endocytosis inhibitor heparin (*Figure 7B,C*).

Further, since both EIPA and CytoD could also inhibit rescue effect of proliferation in BxPC3 and MiaPaCa-2 cells (*Figure 7—figure supplement 1*), suggesting that endocytosis pathway dependent on macropinocytosis and caveolae mediated endocytosis are also associated with uptake of CDEs in pancreatic cancer cells. Additionally, CQ reduced rescue of proliferation by CDEs, thereby suggesting that release of exosomal content through lysosomes may play a role in some cancer cells. Nevertheless, these data suggest that CDEs internalization may happen to various modes of internalization.

To further substantiate the role of KRAS, we used doxycycline inducible *Kras*-G12D cell line (iKras-1, [*Ying et al., 2012*]) to test if enhancement of proliferation by CDEs indeed was KRAS independent. As shown in *Figure 7—figure supplement 2*, iKras-1 cells with or without doxycycline showed similar proliferation increases with CDEs, thereby suggesting that internalization or uptake and supply of exosome-derived metabolites in cancer cells is Kras independent. Clearly, these results suggest that exosomes derived from TME could drive proliferation of PDAC cells by supplying metabolites independent of activated Kras expression.

Previous studies have suggested that Kras can upregulate glycolysis and glutaminolysis. To further evaluate if exosomes mediated metabolic reprogramming is Kras mediated, we measured mitochondrial respiration in PDAC cells with and without pancreatic CDEs. In line with results obtained in prostate cancer cells, we found that OCR of both BxPC3 and MiaPaCa-2 cells were decreased in presence of pancreatic CDEs (*Figure 7D*). Both maximal and reserve mitochondrial capacity of pancreatic cancer cells were significantly reduced in presence of pancreatic CDEs further confirming mitochondrial respiratory capacity inhibition in cancer cells by CAF exosomes (*Figure 7D*). These results suggest that CDEs' action of disabling normal oxidative mitochondrial function is exhibited also in pancreatic cancer and that this regulation is similar in both wild-type (BxPC3) and activated Kras (MiaPaCa-2) expressing cells. Furthermore, there was a corresponding increase in ECAR in both

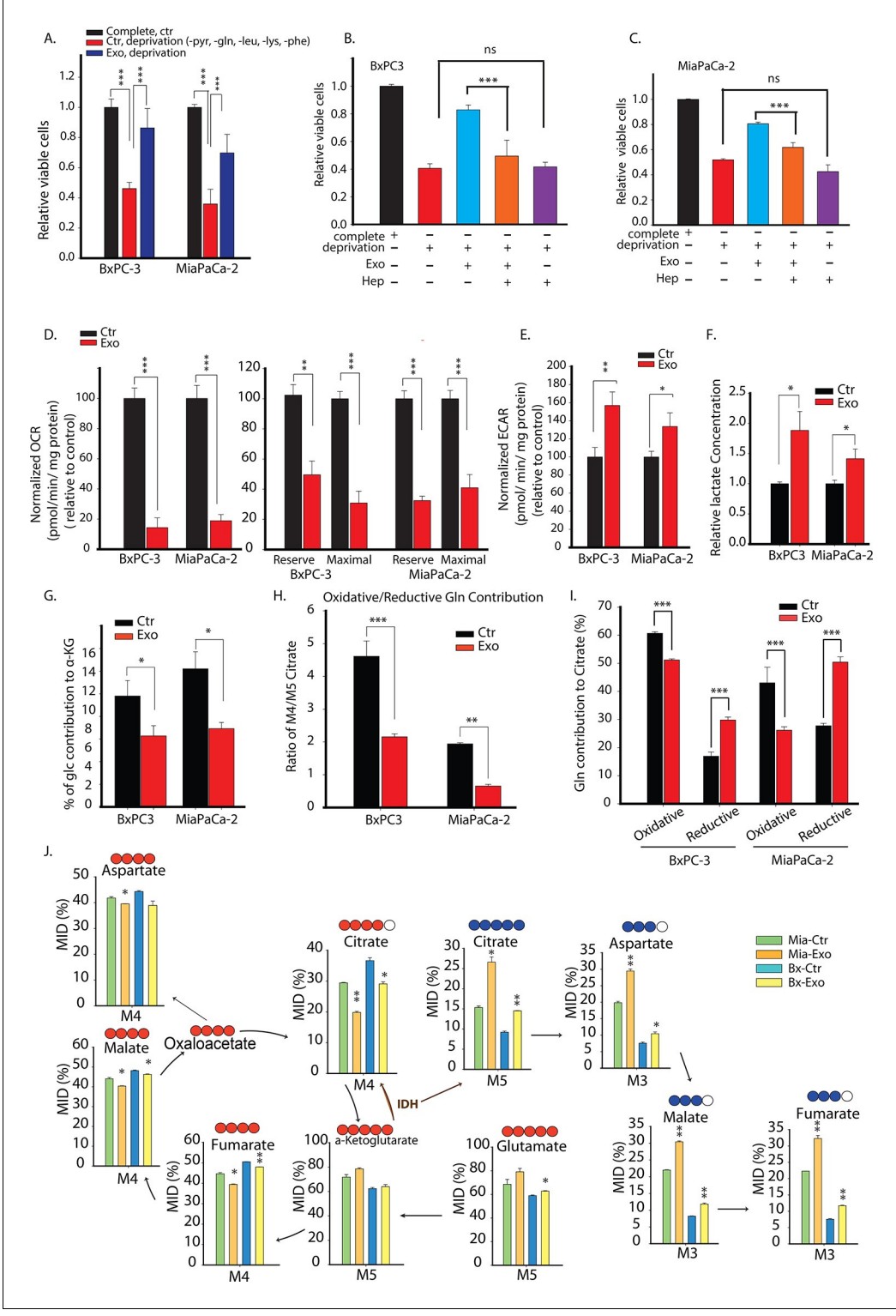

**Figure 7.** Pancreatic CDEs' metabolic reprogramming of pancreatic cancer cells is Kras independent. (**A**) Effect of pancreatic CDEs on pancreatic cancer cell viability under nutrient deprivation (without lys, phe, gln, pyr, leu) conditions. CDEs rescue loss of both wild-type and activated Kras expressing pancreatic cancer cells proliferation under deprivation conditions. Viability of cancer cells with and without exosomes in deprivation condition was measured after 48 hr (n=10). (**B,C**) Heparin inhibit exosomes uptake and thus inhibit the rescue of proliferation by exosomes under nutrient deprived conditions. Heparin (50μg/ml) disrupts receptor-mediated endocytosis. Before adding exosomes, heparin was added to wells for incubation for at least 0.5 hr (n=5). (**D**) Basal OCR were

*Figure 7 continued on next page*

*Figure 7 continued*

measured for BxPC3 and MiaPaCa-2, pancreatic cancer cell lines cultured with pancreatic CAFs (CAF19) exosomes. OCR of both BxPC3 and MiaPaCa-2 were downregulated by CAF19 exosomes. (n=10). Maximal OCR and reserve OCR of BxPC3 and MiaPaCa-2 were downregulated by CAF19 exosomes (n=10). (**E**) ECAR of both BxPC3 and MiaPaCa-2 were upregulated by CAF19 exosomes (n=10). (**F**) Relative lactate abundances were measured using GC-MS in BxPC3 and MiaPaCa-2 cells cultured with and without CAF19-secreted exosomes for 24 hr (n=4). (**G**) Percentage of glucose contribution to α-ketoglutarate in BxPC3 and MiaPaCa-2 cells with and without CAF19-secreted exosomes (n=4). (**H**) Pancreatic CDEs increased reductive glutamine metabolism in wild-type and activated Kras expressing pancreatic cancer cells. Oxidative contribution to citrate is determined by calculating M4 citrate percentage; reductive contribution to citrate is determined by M5 citrate percentage (n=4). (**I**) Ratio of oxidative to reductive glutamine contribution to citrate in wild-type and activated Kras expressing pancreatic cancer cells with CAF19-secreted exosomes (n=4). (**J**) Mass isotopologue distributions (MID) of glutamate, α-ketoglutarate, citrate, malate, and fumarate in BxPC3 and MiaPaCa-2 cancer cells cultured with and without CAF19-secreted exosomes in U-$^{13}C_5$ glutamine (n=4). Higher reductive glutamine metabolism is detected through higher M5 citrate, M3 fumarate, M3 malate, M3 aspartate in pancreatic cancer cells cultured in presence of exosomes. Reductive glutamine metabolism (n=4). Data information: data in (**A**) are expressed as mean ± SD, data in (**B–J**) are expressed as mean ± SEM, *p<0.05, **p<0.01, ***p<0.001. *Figure 7—figure supplements 1–4*.

The following figure supplements are available for figure 7:

**Figure supplement 1.** Effect of drugs inhibiting CDEs uptake and utilization on BxPC3 or MiaPaCa-2 cell proliferation under deprivation (without lys, phe, gln, pyr, leu) conditions.

**Figure supplement 2.** Effect of pancreatic CDEs on pancreatic cancer cell proliferation under nutrient deprivation (without lys, phe, gln, pyr, leu) conditions.

**Figure supplement 3.** Effect of pancreatic CDEs on glycolysis in pancreatic cancer cells using labeled glucose.

**Figure supplement 4.** Effect of pancreatic CDEs on glutamine metabolism in pancreatic cancer cells using labeled glutamine.

pancreatic cancer cell lines in presence of CDEs (*Figure 7E*). This was corroborated by increased lactate levels in the cancer cells in presence of exosomes using U-$^{13}C_6$ glucose labeling based isotope tracer analysis of pancreatic cancer cells in presence of CAF exosomes (*Figure 7F*). Concomitantly, CDEs decreased percentage contribution of glucose to α-ketoglutarate in both pancreatic cancer cell lines (*Figure 7G*, *Figure 7—figure supplement 3*).

To further elucidate the metabolic reprogramming induced by pancreatic CDEs in PDAC cells, we performed GC-MS based isotope labeling experiments using $^{13}C_5$ labeled glutamine. In line with the results obtained in prostate cancers, we found that exosomes from pancreatic CAFs significantly increased the reductive glutamine metabolism (*Figure 7H–J*, *Figure 7—figure supplement 4*). Remarkably, this CAF exosomes-mediated increase of reductive glutamine metabolism was detected in both wild-type and activated Kras expressing pancreatic cancer cells, thus suggesting that metabolic reprogramming induced by stromal exosomes in cancer cells is not only Kras independent but is broadly observed in many cancers.

## Discussion

Altered cell metabolism is one of the hallmarks of cancer. While much of the mechanistic underpinnings in this regard have focused on cell autonomous means of metabolic reprogramming, the objective of this study was to examine the critical role of TME, and in particular, CAFs, might be playing in this adaptive phenomenon. CAFs comprise the majority of cell types within the TME, and their reciprocal interactions with cancer cells have been well documented (*Whiteside, 2008*; *Hazlehurst et al., 2003*; *Karnoub et al., 2007*). Recently, Hu et al. have discovered that CDEs enhance drug resistance in colorectal cancer stem cells by regulating the Wnt pathway and this effect can be reversed by inhibiting exosome secretion (*Hu et al., 2015*). These studies suggest that CDEs may not only enhance cancer growth but may also induce chemoresistance. However, there continues to be sparse data vis-à-vis the role, if any, of the effects CAFs might have on altered

cancer cell metabolism, and the channels of communication that enable such paracrine effects. The objective of our study was to determine whether exosomes released from CAFs might play a role in modulating cancer cell metabolism, allowing neoplastic cells to survive in the nutrient-deprived conditions, a characteristic of many tumors. Our results convincingly demonstrate that not only do exosomes enhance the phenomenon of 'Warburg effect' in tumors, but remarkably, contain *de novo* 'off-the-shelf' metabolites within their cargo that can contribute to the entire compendia of central carbon metabolism within cancer cells.

The Warburg effect, commonly observed characteristic of many cancer types, is identified as the reliance of cancer cells on aerobic glycolysis even under normoxia. This leads to diversion of glucose to lactate thereby creating low pH conditions which modulates TME (*Warburg et al., 1927*; *Gatenby et al., 2006*; *Salimian Rizi et al., 2015*). Although recent studies (*Ishikawa et al., 2008*; *Santidrian et al., 2013*), including our own (*Yang et al., 2014*), implicate mitochondrial activity in cancer metastasis, literature is abound with previous studies based on the hypothesis that aerobic glycolysis may be an outcome of impaired mitochondrial functions. The role of mitochondrial metabolism in tumorigenesis and cancer progression is likely to be organ-specific. Mitochondrial dysfunction could impair oxidative phosphorylation, the TCA cycle, fatty acid oxidation, the urea cycle, gluconeogenesis, and apoptotic pathways (*King et al., 2006*; *Modica-Napolitano and Singh, 2004*). TCA cycle dysfunction leads to oncogenesis by regulating signaling pathway and stabilizing HIF1-$\alpha$ (*Selak et al., 2005*). While many studies in cancer cell metabolism have highlighted higher compensatory glycolysis because of mitochondrial dysfunction; recent studies show that some tumors predominantly use glutamine under hypoxia or conditions mimicking hypoxia such as electron transport chain inhibition through reductive carboxylation for biosynthetic needs (*Yang et al., 2014*; *Wise et al., 2008*). In reductive carboxylation, glutamine is converted to $\alpha$-ketoglutarate, followed by the conversion of $\alpha$-ketoglutarate to isocitrate through isocitrate dehydrogenase (IDH), and isocitrate is converted to acetyl-CoA used for lipid synthesis. The upregulation of the reductive carboxylation pathway enhances cancer cells proliferation (*Metallo, 2012*; *Mullen, 2012*; *Wise et al., 2011*). Although less studied, pyruvate or acetate can act as an alternative source to glucose and glutamine for lipid biosynthesis. Pyruvate is converted to acetyl-CoA through mitochondrial pyruvate dehydrogenase (PDH), whereas acetate is transported into cells and converted to acetyl-CoA through acetyl-CoA synthase (*De Schrijver et al., 2003*; *Feron, 2009*; *Koukourakis et al., 2005*; *Pizer et al., 1996*; *Yoshimoto et al., 2001*; *Zaidi et al., 2012*). Acetyl-CoA is the first step in lipid biosynthesis, which serves biosynthetic needs of proliferating cells. Recent findings implicate TME in the induced metabolic rewiring in cancer cells (*Cairns et al., 2011*; *Fiaschi and Chiarugi, 2012*; *Rattigan et al., 2012*). However, role of exosomes in the metabolic crosstalk between cancer cells and TME is still unknown.

We first asked whether CDEs could reprogram cancer cell metabolism. To determine this metabolic regulation, we first cultured CDEs with prostate cancer cells and performed $^{13}$C based isotope tracing of metabolic fluxes and bioenergetics analysis. We have used 200 µg/ml for the different types of experiments conducted herein (proliferation assays, metabolic assays and tracer experiments). The working concentration was chosen to maintain a physiological ratio of CAFs to cancer cells which is reported to be between 1 and 10 (*Brauer et al., 2013*; *Hu et al., 2015*; *Delinassios, 1987*). A concentration of 200 µg/ml corresponded to ratio between 1 and 5 CAFs per cancer cell and hence, is within the range reported in the literature. However, for different CAFs and cancer cells system this ratio should be individually estimated based on the functional effect of stromal cells on cancer cells. It is to be noted that due to protracted purification steps during exosome isolation, degradation of metabolites can occur and hence, replenishment of exosomes for functional studies was followed and recommended. Additionally, to minimize degradation of metabolites in exosomes, we used fresh exosomes for all the experiments and avoided any freeze-thaw cycle in exosomes that were introduced to cancer cells cultures. Intriguingly, cancer cells cultured with exosomes had significantly reduced OXPHOS with a concomitant increase in glycolysis. Our results were further corroborated by observations regarding higher glucose uptake and lactate secretion by cancer cells in the presence of exosomes. We show for the first time that stromal exosomes shift cellular metabolism towards glycolysis in cancer cells (*Figure 8*). We further extended these observations in prostate cancer to pancreatic cancer and found similar inhibitory effect of pancreatic CDEs on mitochondrial respiration. Interestingly, this regulation was Kras independent in pancreatic cancers. In *Figure 2*, we observe a reduction of OCR in PC3 cells co-transfected with miR-22, let7a and miR-25b, in line with

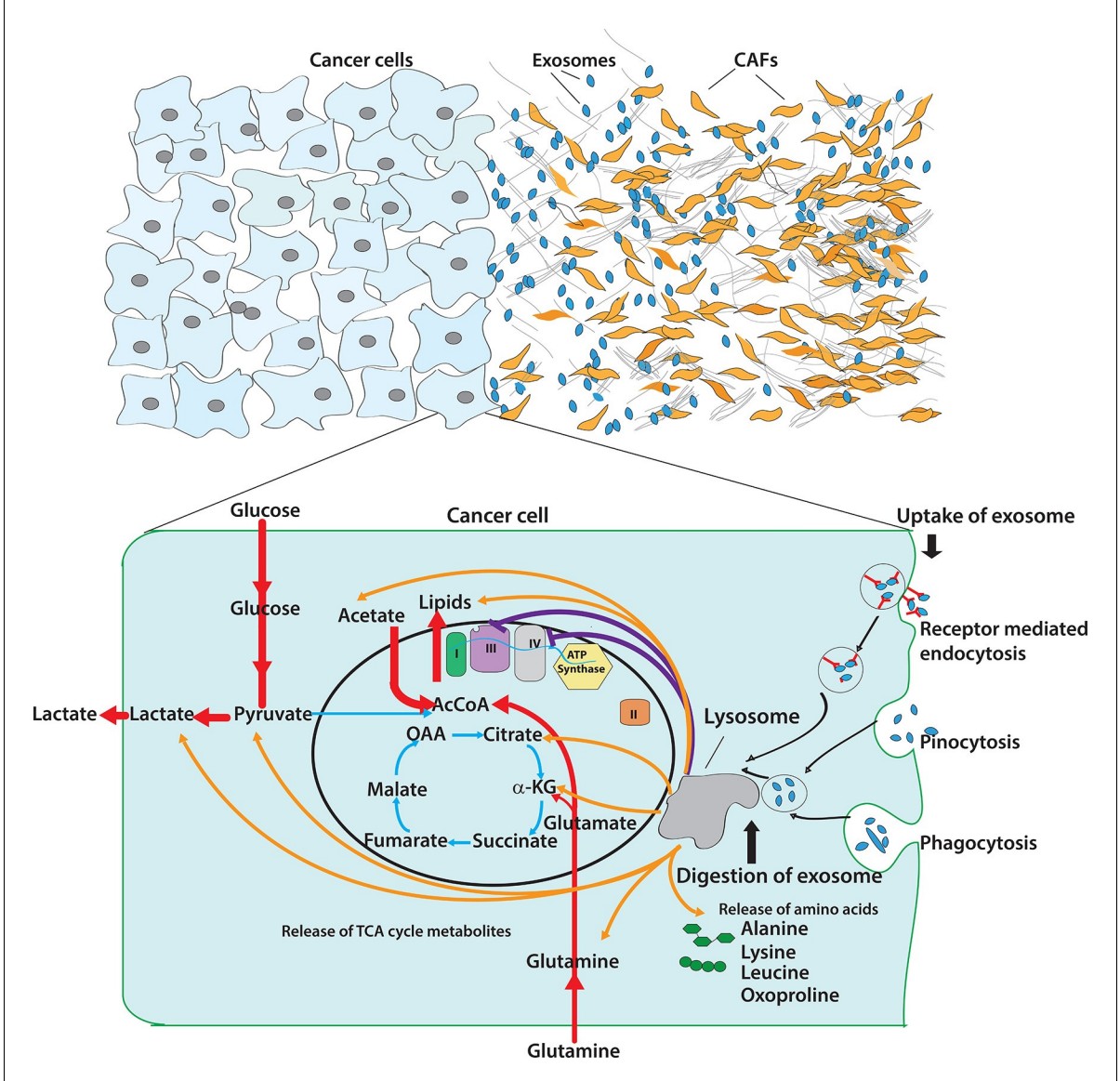

**Figure 8.** Pleiotropic regulation of cancer cell metabolism by CDEs. Schematic shows the metabolic regulation of CDEs in cancer cells through inhibition of oxidative phosphorylation and contribution of metabolite cargo. This regulation leads to significant increase of reductive glutamine metabolism in cancer cells in presence of exosomes. CDEs are also cargo of amino acids, TCA cycle metabolites, and lipids. In nutrient starved TME metabolites derived from exosomes enrich cancer cells with biosynthesis building blocks and thereby promote tumor growth.

our hypothesis of the inhibitory effect of miRNAs from CDEs. However, the caveat associated with this is that due to limitations in genome engineering technologies, it is extremely difficult to insert multiple miRNAs in exosomes or transfect multiple miRNAs together without causing significant toxicity. Nonetheless, our results show that abundant miRNA in exosomes that have been previously shown to target OXPHOS, indeed cause inhibition of oxygen consumption in our system. In light of these results we believe further studies are needed to uncover the mechanism of ETC inhibition by CDEs-derived miRNAs that are out of scope for this study.

Previous studies, reported that under disabled mitochondrial metabolic conditions such as hypoxia, or inhibition of electron transport complexes, cancer cells increasingly rely on reductive glutamine metabolism as compared to oxidative glutamine metabolism. To unravel the contribution of major nutrients to cancer cells, we performed $^{13}$C based metabolite tracing and isotopologue spectral analysis (ISA). Indeed, CDEs upregulated reductive carboxylation of glutamine in cancer cells.

Previous reports showed that reductive carboxylation is a critical pathway to support the growth of tumor cells under hypoxia (*Metallo, 2012*; *Mullen, 2012*). This suggests that CDEs create hypoxia mimicking environment in cancer cells leading to an increase in reductive carboxylation of glutamine in cancer cells. Our ISA results of glucose and glutamine contribution to acetyl-CoA, a precursor for fatty acid synthesis, confirmed the increased reliance of cancer cells on reductive glutamine metabolism in presence of stromal exosomes. However, we were not able to balance palmitate synthesis through glucose and glutamine in cancer cells cultured with CDEs, which led us to measure contributions of acetate and pyruvate. Consistent with recent reports that acetate in media could serve as source of lipid synthesis in cancer cells, we found that acetate could contribute between 10–15% towards lipogenesis. However, pyruvate contribution was much lower and was between 3–8% in cancer cells. These results suggested that exosomes themselves might be acting as source of metabolites in a manner similar to macropinocytosis observed recently in Kras expressing pancreatic cancer cells (*Commisso et al., 2013*). CDE-mediated metabolic changes in cancer cells, metabolic flux analysis is warranted

Since exosomes contained carbon sources such as proteins and lipids, we inquired if exosomes could act as source of building blocks for biosynthesis and proliferation. To our remarkable surprise, we found that exosomes from pancreatic and prostate CAFs contained intact components of the intracellular metabolite pool, including amino acids, acetate, stearate, palmitate, and lactate. We provide previously unidentified evidence that these nutrients can enrich cancer cells under nutrient deprived or nutrient stress conditions. To label the metabolites, proteins and lipids contained in exosomes, we cultured CAFs in media supplemented with $^{13}$C-labeled amino acids dominantly found in the serum (lysine, leucine, phenylalanine, and glutamine) along with nutrients such as glucose and pyruvate. We then cultured cancer cells under nutrient deprived conditions with these labeled exosomes and found that these labeled exosomes could indeed contribute to TCA cycle metabolites in cancer cells. These results conclusively showed that TME can supply metabolites directly to cancer cells through exosomes and these metabolites indeed can fuel TCA cycle in cancer cells. Recently published article by Lyden et al (*Hoshino et al., 2015*), showed that exosomes precondition specific organs for metastatic invasion. Hence, future studies may be directed towards determining organ-specific metabolic reprogramming of CDEs in cancer cells.

Having established that exosomes can fuel TCA cycle in a manner similar to macropinocytosis in prostate cancer, we further showed that this exosomes derived metabolite enrichment is independent of activated Kras expression. Previous studies have shown that tumor cells uptake extracellular nutrients through a mechanism regulated by Kras. From observations made in our pancreatic tumor cell lines, BxPC3 (wild type Kras) and MiaPaCa-2 (activated Kras) we showed that exosomes uptake pathways were independent of Kras expression levels. In our results, BxPC3 and MiaPaCa-2 showed similar extents of metabolic profile regulation as well as enhanced growth rate because of stromal exosomes. These results along with our exosomes uptake inhibition experiments suggest that exosomes uptake occurs in cancer cells through multiple pathways that are independent of activated Kras expression levels. Through endocytosis inhibitors CytoD and receptor mediated endocytosis inhibitor heparin, we inhibited exosomes uptake, and thereby repressed exosomes' influence on increasing growth rates of pancreatic cancer cells BxPC3 and MiaPaCa-2.

In summary, our results reveal insights into intercellular communication between tumor microenvironment and cancer cells. For the first time, we provide evidence that CAFs derived exosomes can reprogram cancer cell metabolism through a metabolite cargo based nutrient enrichment mechanism. Our results will invigorate development of targeted methods for disrupting the exosomes-mediated communication between cancer and stromal cells for in vivo studies and therapeutics based on the targeted inhibition of this crosstalk.

## Materials and methods

### Cells and reagents

PC3, DU145, 22RV1, BxPC3 and MiaPaCa-2 were received from ATCC and authenticated by STR profiling with online ATCC profile. E006AA was kindly povided by Dr. Denis Wirtz (Johns Hopkins University). Patient derived fibroblast cells were kindly provided by Drs. Donna Peehl and Anirban Maitra of Stanford University and MD Anderson, respectively and internal STR profiling is maintained

and checked annually. All cell lines were mycoplasma free based on PCR based assays run every three months in the lab. PKH26 and PKH 67 fluorescent cell linker kits were from Sigma (St. Louis, MO). Exosome–Dynabeads Human CD63 Detection kit (10606D), sheep anti-rabbit IgG Dynabeads (11203D) were from Life technologies (Carlsbad, CA). $^{13}$C carbon-labeled isotopes were from Cambridge Isotope Laboratories (Tewksbury, MA). MiRCURY$^M$ RNA Isolation Kit was from Exiqon (Vedbaek, Denmark). Cytochalasin D was from Sigma. Cell counting kit-8 was from Dojindo (Rockville, MD). 3000 W spin columns were from Life technologies.

Heparin and EIPA were from Sigma. Chloroquine was from VWR (Radnor, PA). Synthetic liposomes (F60103F-DO) were from FormuMax Scientific (Sunnyvale, CA).

## Cell culture

Prostate cancer cells and BxPC3 cells were cultured in RPMI containing 1 mM pyruvate, supplemented with 10% fetal bovine serum (Invitrogen, Carlsbad, CA), 100 U/ml penicillin and 100 U/ml streptomycin. Exosome-depleted FBS (Systems Biosciences, Palo Alto, CA) was used for cell culture when metabolic analysis or proliferation rate measurements were performed. CAF19, CAF35 and MiaPaCa-2 cells were cultured in DMEM. Prostate cancer patient derived fibroblast cells were cultured in MCDB105 (Sigma) supplemented with 5% fetal bovine serum (Invitrogen), 5 ng/ml fibroblast growth factor (FGF) (PeproTech, Rocky Hill, NJ), 5 ng/ml insulin (Sigma), 100 µg/ml gentamicin. All cells were incubated in 5% $CO_2$, and 37°C incubator. CAFs were seeded in T75 flasks, and when the CAFs were 70% confluent, PBS was used to wash cell twice, then the fresh MCDB medium with exosomes depleted FBS (Systems Biosciences) was added to the flask. After 48 hr, exosomes were isolated from the spent medium, and added into the medium incubating prostate cancer cells. For $^{13}$C labeled RPMI medium, we used RPMI without amino acids and supplemented it with appropriate levels of labeled $^{13}$C$_3$-pyruvate, $^{13}$C$_6$-glucose, $^{13}$C$_5$-glutamine, $^{13}$C$_6$-leucine, $^{13}$C$_6$-lysine, $^{13}$C$_9$-phenylalanine; ultracentrifugation was used to remove possible exosomes in FBS of this medium; CAFs were cultured in this medium for 72 hr and labeled exosomes were isolated.

## Exosomes isolation and utilization

To isolate exosomes, cells were cultured with exosome-depleted serum. We collected the conditioned medium to isolate exosomes according to the instructions of the protocol (Life technologies). The collected medium was centrifuged in 2000 xg for 30 min to remove cells and debris. We then transferred the supernatant containing the cell-free culture media to a new tube without disturbing the pellet. Next, we transferred the required volume of cell-free culture media to a new tube and added 0.5 volumes of the Total exosomes isolation (for cell culture media) reagent and mixed the culture media/reagent mixture well by vortexing until there was a homogenous solution. Incubate samples at 2°C to 8°C overnight. After incubation, the samples were centrifuged at 10,000 × g for 1 hr at 2°C to 8°C. The supernatant was aspirated and discarded. Exosomes were contained in the pellet at the bottom of the tube. Re-suspended the pellet in a convenient volume of working medium with exosomes depleted FBS (Systems Biosciences). The concentration of CDEs was measured by BCA kit, which represents the protein concentration of CDEs. The exosome concentration of 200 µg/ml was obtained by diluting an average yield of 270 µg exosome protein (which is equivalent to 5.5x10$^{10}$ particles) which was produced from 120 ml of supernatant. This corresponds to 28000 particles per CAF over a period of 48 hr. Equivalent particle of exosomes was obtained from a measurement of 4.9 µg for 10$^9$ particles.

The utilization of exosomes in cancer cell cultures were based on application of 100–400 µg/ml of exosomes concentration that has been reported in literature (*Christianson et al., 2013*; *Zhu et al., 2012*). A working concentration of 200 µg/ml was chosen for most of the experiments in this study to maintain a physiological ratio of CAFs to cancer cells which is reported to be between 1–10 (*Brauer et al., 2013*; *Hu et al., 2015*; *Delinassios, 1987*). Hence, the number of CAFs required to secrete the amount of exosomes that the cancer cells are exposed to should reflect the ratio of CAFs to cancer cells in tumor. For the different types of experiments conducted herein (proliferation assays, metabolic assays and tracer experiments), this ratio was maintained between 1 and 5 CAFs per cancer cell, and hence is physiologically relevant.

## Exosomes size distribution measurement

Exosomes size and particles density were measured by Zetaview (Particle Metrix, Diessen, Germany). Exosomes resuspended in PBS were diluted 1000 fold for measurement and size distribution. Briefly, 5 µl of exosomes in medium or PBS were added to the measurement system. According to particles' Brownian motion, the diffusion constant is calculated and transferred into a size histogram via the Einstein Stokes relation between diffusion constant and particle size.

## Flow cytometry

Enriched exosomes were captured using the CD63+ Dynabead exosomes isolation kit (Invitrogen, Life Technologies #10606D). The Flow Analysis of stromal exosomes bound to Dynabeads conjugated with antibody was done according to the manufacture's protocol. Briefly, 10 µl of exosomes (200 µg/mL) were incubated with 90 µl of CD63+ Dynabeads overnight at 4°C. Dynabead magnet was then used to positively select for bound exosomes which were then stained with PE Mouse Anti-Human CD63 (BD Bioscience, San Jose, CA). Isotype control was stained by Simultest IgG2a/IgG1 (BD Bioscience, 340394). Flow cytometry was performed on a Accuri C6 System (BD Bioscience) and analyzed on Flow Jo software.

To analyze exosomes uptaken by prostate cancer cells, exosomes were pre-labeled by PKH67 dye (Sigma), and 3000 spin columns were used to remove extra dye. The dyed exosomes were added to RPMI medium to culture cancer cells for 3 hr and then flow cytometry was performed to measure fluorescence intensity of cells.

## Fluorescence microscopy to image exosomes uptake by prostate cancer cells

Exosomes were pre-labeled according to PKH26 cell linker kit (Sigma). 3000 spin columns (Sigma) were used to remove extra dye. PC3 Cells were grown to 50% confluence in 8-well chamber slides and incubated with PKH26 labeled exosomes (200 µg/mL) for 3 hr. Cells were then washed two times with PBS solution and fixed with 4% PFA for 10 min. Nuclei were stained with 4', 6-diamidino-2-phenylindole (DAPI) and slides were viewed under a Axio Observer Z1 Inverted fluorescence microscope (Zeiss) and analyzed on Zen software.

## Viability assay

Cells viability was measured by Cell counting kit-8 (Dojindo Molecular Technologies, Inc., Rockville, MD). Cells were cultured on 96-well plate in the indicated conditions. Viability assay solution was added to the plate for incubation of 3 hr and absorbance was measured at 450 nm.

## RNA purification and amplification for Illumina Microarrays

Total RNA was extracted from cells using the Quick-RNA MiniPrep (Zymo Research, Irvine, CA), following the manufacturer's instructions. RNA amplification was performed using Illumina Total-PrepTM RNA amplification kit (AMIL1791, Life Technologies), according to the manufacturer's instructions. Briefly, 500 ng total RNA was used to synthesize the first strand cDNA using a MyCycler thermal cycler (Bio-Rad, Hercules, CA). Subsequently, the second strand cDNA was synthesized and cDNA was purified with 20 µl of 55°C nuclease-free water. In vitro transcription for cRNA synthesis was carried out using 14 hr incubation at 37°C. cRNA was then eluted with 200 µl of 55°C nuclease-free water. Hybridization and imaging were done using the HumanHT-12 v4 Expression BeadChip Kit (Illumina, San Diego, CA) according to manufacturer's protocol.

## Analysis of gene expression using real-time PCR

Total RNA was isolated using a Zymo mini kit (Qiagen, Valencia, CA). High Capacity cDNA Reverse Transcription Kit (Applied Biosystems, Foster City, CA) was used to synthesize cDNA from 1 µg of total RNA. The levels of COX-1 and CYTB were examined by real-time PCR using 50 ng of the synthesized cDNA. Real-time PCR was performed with the SYBR Green PCR MasterMix (Applied Biosystems, Warrington, UK). All reactions with COX-1 and CYTB were normalized against glyceraldehyde-3-phosphate dehydrogenase (GAPDH). Specific primer sets were as follows (listed 5'–3'; forward and reverse, respectively): COX-1, TCGCATCTGCTATAGTGGAG and ATTATTCCGAAGCCTGGTAGG;

CYTB, TGAAACTTCGGCTCACTCCT and AATGTATGGGATGGCGGATA. Reactions were performed in a volume of 20 µl.

## miRNA measurements from exosomes

Isolation of miRNA from exosomes was done with the MiRCURY RNA Isolation Kit. In brief, steps of lysis, precipitation, repeated washing, and elution were performed to isolate purified small RNAs and then miRNA expression levels were measured by NanoString miRNA assays. miRNA levels were measured using the nCounter Human V2 miRNA expression analysis kit (Nanostring), according to the manufacturer's instructions. The data were corrected for loading using the relative geometric means of endogenous miRNA levels as a correction factor. The miRNAs were ranked by their average count across all exosomes samples.

## Glucose assay

Glucose assay were done according to the instructions of assay kit (Wako Glucose kit, Wako Diagnostics, Mountain View, CA). In brief, a 250 µl of reconstituted Wako glucose reagent was added to a 96-well assay plate followed with 2 µl sample addition in each well. The plate was incubated at 37°C for 5 min. The change in absorbance, which indicates the amount of glucose present, was measured at 505 nm and 600 nm by using a spectrophotometer (SpectraMax M5; Molecular Devices, Sunnyvale, CA).

## Lactate assay

Lactate secretion was determined using the Trinity Lactate Kit (Trinity Biotech Plc., Co Wicklow, Ireland). Media samples were diluted 1:10 in PBS, and lactate reagent was reconstructed and added to the diluted samples in an assay plate. The plate was incubated for 1 hr at 37°C, protecting from light. Afterwards the change in absorbance was read on a spectrophotometer at 540 nm.

## Protein assay

Protein assays are used to do normalization in our experiment and is done according to Bicinchoninic Acid Protein Assay (Thermo Fisher Scientific, Waltham, MA) protocol. In brief, protein reagent was added to a 96-well assay plate and mix with samples or standard, and then incubated at 37°C for 30 min. The absorbance was read on a spectrophotometer at 562 nm.

## Acetate assay

Acetate concentration was measured according to manufacturer's instructions for acetate colorimetric assay kit (BioVision #K658, Milpitas, CA). Briefly, samples or acetate standards were mixed with reaction mixtures, incubated at room temperature for 40 min, and measured at $OD_{450 nm}$.

## Measurement of mitochondrial membrane potential

Mitochondrial permeability transition was determined by staining the cells with TMRM (Molecular Probes, Eugene, OR). The mitochondrial membrane potential was quantified by SpectraMax M5 (SpectraMax M5; Molecular Devices, Sunnyvale, CA).

## Measurements of oxygen consumption rate and extracellular acidification rate

Mitochondrial oxygen consumption was monitored with an XF24 Extracellular Flux Analyzer (Seahorse Bioscience, North Billerica, MA). The cells were seeded in Seahorse 24-well microplates at a cell density of 70% confluent cells per well in 100 µL of culture media with indicated conditions. After overnight incubation at 37°C with 5% $CO_2$, the media was replaced with 700 µL of assay medium. Then incubate the plate at 37°C without $CO_2$ for at least 1 hr. The oxygen consumption rate (OCR) was then measured. The endogenous coupling degree of the OXPHOS system was assessed using oligomycin (2 µg/ml), an inhibitor of the $F_1F_O$-ATPsynthase. The uncoupled OCR was also measured in presence of 2.5 µM of FCCP. Finally, the cells were treated with a mitochondrial complex I inhibitor, Rotenone (2 µM) in order to assess the mitochondrial contribution to OCR. Extracellular acidification rate (ECAR) can be measured in a similar way to OCR. All OCR or ECAR value was normalized with protein content of cells.

## Isotope labeling analysis using GC-MS

### Metabolites extraction

Cancer cells were seeded in 6-well plates overnight, and replaced with medium containing U-$^{13}C_6$ glucose or U-$^{13}C_5$ glutamine. After 24/48/72 hr, medium was aspirated, and cells were washed with cold PBS once and quenched with 400 µl of cold methanol. Same volume of water containing 1 µg of norvaline (internal standard) was added, and cells were scraped into Eppendorf tubes. 800 µl of chloroform was added into the tubes, and vortexed at 4°C for 30 min, centrifuged at 7300 rpm for 10 min at 4°C. The aqueous layer was collected for metabolite analysis and the chloroform layer was collected for fatty acids analysis.

### Derivatization

Aqueous samples were dried and dissolved in 30 µl of 2% methoxyamine hydrochloride in pyridine (Pierce, Waltham, MA), and sonicated for 10 mins. Afterwards, samples were kept in 37°C for 2 hr. Samples were kept for another 1 hr at 55°C after addition of 45 µl of MBTSTFA+1% TBDMCS (Pierce). Chloroform samples were dried and dissolved in 75 µl Methyl-8 Reagent (Pierce), and incubate at 60°C for 1 hr. Samples were transferred into vials containing 150 µl of insert (Thermo Fish Scientific).

### GC/MS measurements

GC/MS analysis was performed using an Agilent 6890 GC equipped with a 30-m Rtx-5 capillary column for metabolites samples or 30 m DB-35 MS capillary column for fatty acids samples, connected to an Agilent 5975B MS. For metabolites samples, the following heating cycle was used for the GC oven: 100°C for three minutes, followed by a temperature increase of 5°C/min to 300°C for a total run time of 48 min. For fatty acids samples, the following heating cycle was used for the GC oven: 100°C for 5 min increased to 200°C at 15° min$^{-1}$, then to 250°C at 5° min$^{-1}$ and finally to 300°C at 15° min$^{-1}$. Data was acquired in scan mode. The abundance of relative metabolites was calculated from the integrated signal of all potentially labeled ions for each metabolite fragment.

### Intra-exosomal metabolites extraction

The exosomes pellet was extracted by adding 75 µl of cold methanol; 150 µl of cold water (with norvaline for GC-MS measurement, without norvaline for UPLC measurement) was added, which dissolved exosomes completely. 20µl of the liquid was stored for protein assay. Then 150 µl of cold chloroform was added into the tubes and vortexed at 4°C for 30 min, centrifuged at 7300 rpm for 10 min at 4°C. The aqueous layer was collected for intra-exosomal metabolite analysis. Chloroform layer was stored for lipid analysis.

## Statistical analysis

The results presented are expressed in mean value of N experiments ± S.D or SEM, with N≥2, n≥ 3. Comparison of the data sets obtained from the different experiment conditions was performed with the Student t test. In the bar graphs, single asterisk (*) represents p<0.05, double asterisks (**) represent p<0.01 and triple asterisks (***) represent p<0.001.

## Acknowledgements

This work made possible in part through support from the Ken Kennedy institute for Information technology at Rice University to DN under the Collaborative Advances in Biomedical Computing 2011 seed funding program supported by the John and Ann Doerr Fund for the Computational Biomedicine.

## Additional information

### Funding

No external funding was received for this work.

## Author contributions

HZ, Conception and design, Acquisition of data, Analysis and interpretation of data, Drafting or revising the article; LY, JB, AA, VB, TM, JCM, SG, PTR, AM, Acquisition of data, Analysis and interpretation of data, Drafting or revising the article; TT, EGS, FASL, HA, Acquisition of data, Analysis and interpretation of data, Contributed unpublished essential data or reagents; SNM, LC, Acquisition of data, Analysis and interpretation of data; DP, Acquisition of data, Drafting or revising the article, Contributed unpublished essential data or reagents; DN, Conception and design, Analysis and interpretation of data, Drafting or revising the article

## Author ORCIDs

Deepak Nagrath, http://orcid.org/0000-0002-8999-2282

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
