## [Decision Letter]

Thank you for submitting your work entitled "Tumor Microenvironment Derived Exosomes Pleiotropically Modulate Cancer Cell Metabolism" for consideration by *eLife*. Your article has been favorably evaluated by Charles Sawyers (Senior editor) and three reviewers, one of whom, Chi Dang, is a member of our Board of Reviewing Editors.

The reviewers have discussed the reviews with one another and the Reviewing editor has drafted this decision to help you prepare a revised submission. Please note that many of the essential revisions are clarifications of data interpretation and acknowledgement of caveats and should not require lengthy additional experiments.

Summary:

The manuscript by Zhao et al. reports that cancer associated fibroblast(CAF)-derived exosomes (CDEs) changes epithelial cancer cell metabolism. The authors used CAFs from 4 prostate cancer cell lines, a number of primary prostate cancers, as well as 2 pancreas cancer cell lines and 2 primary pancreas cancers, to show that CDEs contain an array of metabolites including amino acids, TCA cycle metabolites, fatty acids and precursors/intermediates. Delivery of these exosomes to cancer cells inhibited oxidative phosphorylation (although this in turn leads to a series of downstream sequela including increased glycolysis, TCA reductive carboxylation, etc. through metabolic labeling using 13C-glucose or glutamine). By examining gene expression profiles, the authors found a decrease in OXPHOS gene expression in prostate cancer cells exposed to CDEs. Quantitative analysis suggests that 5-29% of various prostate tumor metabolites were derived from 13C-labeled CDEs. Extending these observations, the authors demonstrate that pancreatic cancer cells could also take up CDEs via a macropinocytosis mechanism that appears independent of K-ras mutational status. A number of major issues, if addressed, would make this manuscript meritorious for publication.

Essential revisions:

1) The main missing component of the paper is the mechanism of oxidative phosphorylation inhibition. This is hinted at in Figure 8—figure supplement 1. However, any mechanism acting through transcription would account for such a strong effect over only 24 h (oxidative phosphorylation proteins do not typically turnover so quickly). How can an increase of the intracellular metabolite pool by CDEs of less than 5% for most metabolites (Figure 6) lead to such marked phenotypes in terms of growth and Glycolysis/OXPHOS? As the authors acknowledge in their Introduction, miRNAs are an important cargo of exosomes. The authors have found their CDEs to be enriched in several miRNAs targeting OXPHOS enzymes (Figure 8—figure supplement 1). These observations are in line with the presented hypothesis and it could explain how CDEs can decrease OXPHOS in cancer cells. Why is this point made "secondary", by showing some data in the last supplemental figure and not referring to it until one of the last discussion chapters? The reviewer believes that this set of data should be highlighted and possibly further explored in the manuscript.

2) Could the microsomes have an effect on tumor cell metabolism independent of the microsomal metabolite content? As a rigorous control for OXPHOS effects of microsomes, synthetic liposomes with a size distribution of microsomes should be used to rule out a metabolic response to liposome uptake that is independent of the contents of the microsomes.

3) Throughout the manuscript, authors use 200ug/mL of isolated exosomes for the different experiments based on the observation that this is the maximal up-taken concentration by cancer cells (Figure 5). How does this value compare to what stromal cells can secrete? In other words, how physiological are the 200ug/ml of isolated CDEs? This is a very important point as it goes to what may actually be happening in the tumor itself.

4) While many of the tracer results are internally consistent (e.g. labeling of glutamate typically matches that of α-Ketoglutarate), the totality of the results are probably not easily quantitatively explainable with current metabolic maps. The authors should acknowledge that they are not doing flux identification and that some elements of the data seem inconsistent, e.g. too much M4 citrate relative to M3/M4 KG/Glu in Figure 4 more specific concern regards the claims about the oxidative pentose phosphate pathway. There are more appropriate tracers for this purpose and the amount of PPP overflow is not enough to typically read out with the type of method they are using. Their data are indeed a little puzzling, particularly why there is so much more M0 than M1 lactate in all of their conditions, or why it changes with exosomes, but assuming that they do not want to investigate this thoroughly, they should acknowledge the oddity but not blame it on the PPP.

Do total pools of metabolites, other than lactate (Figure 3) change significantly in cancer cells treated with CDEs? Are the total levels of TCA cycle metabolites (Figure 3) or amino acids (Figure 6), changed? Authors should also represent total ion currents in addition to the fractional labeling representation as it provides further information and would help answer these questions.

5) A limitation of this manuscript is the metabolomics on the CDEs themselves. The isolation of exosomes is quite protracted, opening up the possibility of artifactual changes in the metabolome. As the authors know, this is why the isolation of metabolites is extremely time sensitive and all proteins are denatured immediately with such reagents as ice cold methanol. This may be technical limitation of exosome isolation that could severely impact the results. This issue needs addressing. The quantitation of metabolites in the exosomes leads to some astronomical values per unit of protein. Perhaps this is because the exosomes are very low in protein? In a typical cell, which may be about 20% protein, the concentration of lactate of 50 umol/mg prot would correspond to a molar concentration of 10 M! The authors should carefully check this.

6) The main metabolite component of the exosomes seems to be Gln (Figure 5). This amino acid is a major source of carbon for the TCA cycle in different systems. Also, in Figure 6—figure supplement 1, the major labeled citrate isoform found in PC3 cells after treatment with "labeled" CDEs is M2 and not M6 – suggesting that the majority of this citrate is not coming directly from the exosome labeled citrate but rather from Acetyl-CoA (or precursors) delivered by exosomes, further suggesting feeding of the TCA cycle by exosome delivered metabolites – how do the authors reconcile this potentially higher TCA cycle activity with the decreased OXPHOS phenotype?

7) The authors do not elucidate how the tumor cells uptake the nutrient loaded exosomes. The effects of heparin and CytoD should be shown also on the control condition, to ensure that things like proliferation inhibition are specific to the deprivation/restoration with exsomes. The use of macropinocytosis inhibitors (EIPA) should provide an answer to this question. If the delivery of endosomal contents in the cell requires degradation of exosomes by fusion with lysosome (to release their cargo) in cancer cells, then treatment with drugs that block this process such as CQ or BafilomycinA should impair the phenotypes observed with cancer cells are treated with CDEs.

8) One of the sections points at excluding KRAS from playing a role in PDAC uptake of CDEs. However, to fully exclude KRAS role, the analysis should be based on more than comparison between two cell lines. For example, if the phenotype (decreased OCR or increased proliferation upon treatment with CDEs) is maintained after genetic ablation of KRAS in MiaPaCa cells.

9) The mitochondrial membrane potential is decreased in PC3 cells upon treatment with CDEs (Figure 2), did the authors verify if total mitochondrial content and/or shape is altered in cancer cells treated with CDEs? Also, did the authors verify if CDEs contain mitochondria or fractions of mitochondria? This could explain the relatively high levels of malate and other TCA cycle intermediates found inside (Figure 5).

[Editors' note: further revisions were requested prior to acceptance, as described below.]

Thank you for resubmitting your work entitled "Tumor Microenvironment Derived Exosomes Pleiotropically Modulate Cancer Cell Metabolism" for further consideration at *eLife*. Your revised article has been evaluated by Charles Sawyers (Senior editor), a Reviewing editor, and two reviewers. The manuscript has been improved. However, there are significant remaining issues that we would like for you to provide a response, within 1-2 weeks, on how you would address them to determine whether a second revision is invited.

While the effect of CAF-derived exosomes on OXPHOS (Figure 7) is impressive, the mechanisms by which this occurs remain undefined (issue 1, below), particularly with the issues surrounding the exosome metabolome (issue 2, below).

1) The miRNA data is still very correlative and quite weak (based on literature showing that particular miRNAs can regulate expression of various oxphos enzymes).

2) The concentrations of metabolites in the exosomes continue to be unrealistic, e.g. 1 moles glutamine, 5M lactate moles, per g protein, which would translate to (assuming exosomes are 20% protein), 200M and 1000M, which is physically impossible. What is the relationship between exosome protein, exosome volume, and exosome number? Complete analytical support is necessary, including raw data, for any metabolites that the authors claim are present in exosomes at concentrations exceeding 20 mM.

In addition to the two major issues above, there are other items that need addressing.

1) Figure 6 seems confusing. Since lysine is an essential amino acid, why are they seeing M+2 form? Why do lactate, pyruvate, and alanine, which all share the same carbon skeleton, all have different labeling patterns? How do they get to the conclusion that ~ 30% of KG is derived from exosome metabolites? It does not seem proper to normalize the KG labeling in cells to that in exosomes, when most KG in cells comes from glutamine or glutamate. It is also confusing how labeling of KG in cells would exceed glutamine and glutamate. I do not want to cause the authors excessive suffering on this point. To this end, they may be better served reporting less isotope labeling data, instead of reporting dubious data. The leucine data, for example, makes the point that the exosome contribution is modest while looking more straightforward than some other data. Similarly, it seems important to see results for glutamine. Perhaps a few well-chosen examples would suffice. Also, perhaps some funny labeling patterns can be cleaned up by more careful examination of the raw data to eliminate interfering peaks.

2) The authors provide some more rationale for choosing the concentration of exosomes used in their studies. However, there is still uncertainty as to how this represents the physiological state. The authors should at least write a sentence or 2 in the Discussion talking about this limitation. The limitation of this analysis (protracted purification protocol) should be mentioned in the Discussion.

3) The correction of the lactate levels to 1/10th what was originally reported does not seem "minor". Can the authors please explain this further?

4) It is still not clear why M+2 is the dominant citrate isotopomer in Figure 6—figure supplement 1. This continues to suggest that the majority of this citrate is not coming directly from the exosome labeled citrate but rather from Acetyl-CoA (or precursors) delivered by exosomes.

5) In the bar graphs looking at drug effects, the authors should not limit tests of statistical significance to the hypothesized effects (in the exosome rescue conditions), but also conduct the same tests in the non-rescue conditions, as it seems that the drugs sometimes have an effect also in those conditions. If those tests were all non-significant, there is no need to mark them on the figure, but that should be clearly reported.

6) The authors should be aware that oxoproline is most commonly found due to degradation of glutamate during sample processing. They should not make main text statements relating this to the glutathione pathway without further evidence.

---

## [Author Response]

*1) The main missing component of the paper is the mechanism of oxidative phosphorylation inhibition. This is hinted at in Figure 8—figure supplement 1. However, any mechanism acting through transcription would account for such a strong effect over only 24 h (oxidative phosphorylation proteins do not typically turnover so quickly). How can an increase of the intracellular metabolite pool by CDEs of less than 5% for most metabolites (Figure 6) lead to such marked phenotypes in terms of growth and Glycolysis/OXPHOS? As the authors acknowledge in their Introduction, miRNAs are an important cargo of exosomes. The authors have found their CDEs to be enriched in several miRNAs targeting OXPHOS enzymes (Figure 8—figure supplement 1). These observations are in line with the presented hypothesis and it could explain how CDEs can decrease OXPHOS in cancer cells. Why is this point made "secondary", by showing some data in the last supplemental figure and not referring to it until one of the last discussion chapters? The reviewer believes that this set of data should be highlighted and possibly further explored in the manuscript.*

We thank the reviewers for pointing this out and believe that the reviewers’ observation is correct in this regard. Based on the suggestion, we have moved the miRNA data from supplementary figure to the main Figure 2. The miRNA literature suggests that often times miRNAs act in clusters, however there are inherent technological limitations in co-transfecting multiple miRNAs into our exosome-based cell assays as we found these to be toxic to the cells. Based on our initial observations and technical difficulties, we realized that it was beyond the scope of the current work and hence, we had included the miRNA exosomal abundance data in supplementary figure in our previous version. Please see the corrections below, which were added in the revised manuscript.

We have moved the following text from Discussion to text. Subsection “CDEs downregulate mitochondrial function of prostate cancer cells”, last paragraph: It is well established that exosomes contain noncoding RNAs (e.g. miRNAs) which can serve as a communication mechanism between stromal and cancer cells. […] In summary, these results suggest that CDEs reduced mitochondrial oxidative phosphorylation and induced metabolic alterations in cancer cells mimicking hypoxia-induced alterations.

We have added the following new discussion in connection with the above observations. Discussion, third paragraph:“Further studies are needed to uncover the mechanism of ETC inhibition by CDEs-derived miRNAs.”

*2) Could the microsomes have an effect on tumor cell metabolism independent of the microsomal metabolite content? As a rigorous control for OXPHOS effects of microsomes, synthetic liposomes with a size distribution of microsomes should be used to rule out a metabolic response to liposome uptake that is independent of the contents of the microsomes.*

We thank the reviewers and editors for bringing this important control to our attention. To verify the effect of the metabolite content, we used synthetic liposomes [DOPC/CHOL (DOPC: 1,2-dioleoyl-sn-glycero-3-phosphocholine, CHOL: cholesterol) liposomes labeled with DiO, size 85-110nm] and cultured cancer cells with and without liposomes. The liposomes used were of similar size distribution as exosomes. We measured prostate and pancreatic cancer cells’ proliferation, oxygen consumption rate (OCR), and extracellular acidification rate (ECAR) with and without synthetic liposomes. Our results show that cancer cells indeed uptake synthetic liposomes (Figure 2—figure supplement 2). However, liposomes neither enhanced cancer cells proliferation or ECAR, nor inhibited OCR. Nevertheless, this was an important control indeed and it allowed us to conclusively link the CDEs induced changes in cancer cell metabolism and growth rate to metabolic content. We have updated our manuscript with these control results and made corrections as indicated below. The description of liposome results has been added on page 6, line 2 and Figure 2—figure supplement 2.

Subsection “CDEs downregulate mitochondrial function of prostate cancer cells“, second paragraph: “To conclusively associate CDEs induced metabolic reprogramming with metabolic content of exosomes, we verified if synthetic liposomes (DOPC/CHOL liposomes labeled with DiO, size 85-110nm; DOPC: 1,2-dioleoyl-sn-glycero-3-phosphocholine, CHOL: cholesterol) with a size distribution similar to exosomes could similarly modulate cancer cells. Our data suggests that liposomes did not alter cell proliferation, OCR and ECAR in both prostate (PC3) and pancreatic cancer cells (MiaPaCa-2 and BxPC3) (Figure 2—figure supplement 2). These results implicate metabolic content of exosomes towards observed changes in CDEs-induced increased cancer cell proliferation, mitochondrial dysfunction and increased glycolysis.”

*3) Throughout the manuscript, authors use 200ug/mL of isolated exosomes for the different experiments based on the observation that this is the maximal up-taken concentration by cancer cells (Figure 5). How does this value compare to what stromal cells can secrete? In other words, how physiological are the 200ug/ml of isolated CDEs? This is a very important point as it goes to what may actually be happening in the tumor itself.*

We thank the reviewers and editors for pointing this out. Based on the application, 100-400 μg/ml of exosomes concentration has been reported in literature (Christianson et al., 2013; Zhu et al., 2012). A working concentration of 200 μg/ml was chosen for all the experiments in this study to maintain a physiological ratio of CAFs to cancer cells which is reported to be between 1-10 (Brauer et al., 2013; Delinassios, 1987; Hu et al., 2015). Hence, the number of CAFs required to secrete the amount of exosomes that the cancer cells are exposed to should reflect the ratio of CAFs to cancer cells in tumor. For the different types of experiments conducted herein (proliferation assays, metabolic assays and tracer experiments), this ratio was maintained between 1 and 5 CAFs per cancer cell, and hence is physiologically relevant. We have also clarified this in the Methods section of the manuscript as written below.

Methods, subsection "Exosome isolation and utilization":

“The utilization of exosomes in cancer cell cultures were based on application of 100-400 μg/ml of exosomes concentration that has been reported in literature (Christianson et al., 2013; Zhu et al., 2012). A working concentration of 200 μg/ml was chosen for all the experiments in this study to maintain a physiological ratio of CAFs to cancer cells which is reported to be between 1-10 (Brauer et al., 2013; Delinassios, 1987; Hu et al., 2015). Hence, the number of CAFs required to secrete the amount of exosomes that the cancer cells are exposed to should reflect the ratio of CAFs to cancer cells in tumor. For the different types of experiments conducted herein (proliferation assays, metabolic assays and tracer experiments), this ratio was maintained between 1 and 5 CAFs per cancer cell, and hence is physiologically relevant.”

*4) While many of the tracer results are internally consistent (e.g. labeling of glutamate typically matches that of α-Ketoglutarate), the totality of the results are probably not easily quantitatively explainable with current metabolic maps. The authors should acknowledge that they are not doing flux identification and that some elements of the data seem inconsistent, e.g. too much M4 citrate relative to M3/M4 KG/Glu in Figure 4. A more specific concern regards the claims about the oxidative pentose phosphate pathway. There are more appropriate tracers for this purpose and the amount of PPP overflow is not enough to typically read out with the type of method they are using. Their data are indeed a little puzzling, particularly why there is so much more M0 than M1 lactate in all of their conditions, or why it changes with exosomes, but assuming that they do not want to investigate this thoroughly, they should acknowledge the oddity but not blame it on the PPP.*

*Do total pools of metabolites, other than lactate (Figure 3) change significantly in cancer cells treated with CDEs? Are the total levels of TCA cycle metabolites (Figure 3) or amino acids (Figure 6), changed? Authors should also represent total ion currents in addition to the fractional labeling representation as it provides further information and would help answer these questions.*

We completely agree with reviewers that we performed isotope tracing analysis and did not estimate metabolic fluxes. We have added a statement in Discussion section regarding this issue.

We are hoping that the question that reviewers had about M4 citrate is referred to Figure 4. Since Figure 4 describes experimental resultsobtained by using uniformly-labeled glutamine, levels of M4 citrate derived from oxidative metabolism of glutamine do correspond with M5 a-KG/Glu. Also, as is mentioned in the text and seen by previous researchers, mitochondrial ETC inhibition displays a response similar to hypoxia and thus increases glutamine-driven reductive carboxylation in cancer cells with CDEs. However, the reductive glutamine’s metabolism contributes to M5 citrate from M5 a-KG. The levels of M5 citrate are lower than M5 a-KG/Glu in our results.

We agree with reviewers that our tracing cannot clearly resolve pentose phosphate pathway and hence in the revised manuscript, we have removed that statement. The reason for high M0 lactate is because of pyruvate in the media, which results in dilution of labeled lactate.

We have included the total ions current for TCA cycle metabolites in Figure 3—figure supplement 1. As seen in Figure 3—figure supplement 1, there is an increase in lactate, pyruvate, and TCA cycle metabolites in cancer cells with CDEs. As discussed in the text, low citrate levels could be explained because of reduced glucose contribution to TCA cycle in the presence of exosomes. It is also to be noted that in Figure 4, we showed that in cancer cells cultured with CDEs there are other precursors of acetyl-CoA, which were found to maintain lipogenesis. There were non-significant changes in levels of amino acids in cancer cells with and without CDEs, which could be because of their utilization for nucleotides and protein synthesis.

Subsection “CDEs upregulate glucose metabolism in cancer cells“, second paragraph: “We found that the CDEs increased the lactate levels in the cancer cells (Figure 3, Figure 3—figure supplement 1).”

*5) A limitation of this manuscript is the metabolomics on the CDEs themselves. The isolation of exosomes is quite protracted, opening up the possibility of artifactual changes in the metabolome. As the authors know, this is why the isolation of metabolites is extremely time sensitive and all proteins are denatured immediately with such reagents as ice cold methanol. This may be technical limitation of exosome isolation that could severely impact the results. This issue needs addressing. The quantitation of metabolites in the exosomes leads to some astronomical values per unit of protein. Perhaps this is because the exosomes are very low in protein? In a typical cell, which may be about 20% protein, the concentration of lactate of 50 umol/mg prot would correspond to a molar concentration of 10 M! The authors should carefully check this.*

We agree with the reviewers that technical limitations of the exosome isolation protocol could impact the metabolomic analysis of exosomes. We were cognizant of this issue and tried to minimize steps, which could cause degradation of the metabolites. However, our data clearly supported both (i) phenotypic changes observed by rescue of viability during nutrient deprivation conditions and (ii) metabolic contributions observed through isotopically enriched metabolites in cancer cells through labeled CDEs. Since the results were consistent for different experimental repeats and also across cell lines, we feel confident about the conclusions. Further, as seen in Figure 6—figure supplement 1, the percentage contribution of labeled exosomes (estimated as mean enrichment normalized to exosome enrichment) is significant. This particular result could not have been impacted by isolation procedures. It is more likely that we are underestimating and not overestimating the levels of these metabolites.

We agree with the reviewer that low protein content in exosomes could lead to apparently higher concentrations. However, this is the best possible estimate of concentrations as is done with cellular or organelle level normalizations. We would like to point that we have made a minor correction in lactate concentration, which was because of range of standards used over the period of time. Currently lactate concentrations are 1/10_th_ of the previous values.

*6) The main metabolite component of the exosomes seems to be Gln (Figure 5). This amino acid is a major source of carbon for the TCA cycle in different systems. Also, in Figure 6—figure supplement 1, the major labeled citrate isoform found in PC3 cells after treatment with "labeled" CDEs is M2 and not M6 – suggesting that the majority of this citrate is not coming directly from the exosome labeled citrate but rather from Acetyl-CoA (or precursors) delivered by exosomes, further suggesting feeding of the TCA cycle by exosome delivered metabolites – how do the authors reconcile this potentially higher TCA cycle activity with the decreased OXPHOS phenotype?*

We thank the reviewers for bringing forward this interesting point. Our results show that CDEs induce decreases of OXPHOS activity, which upregulates glutamine-driven reductive carboxylation. The upregulated reductive glutamine metabolism maintains higher citrate synthesis for lipogenesis and cell proliferation. However, as reviewer correctly pointed out acetyl CoA may also contribute to higher M2 citrate levels. Our data suggest that both reductive carboxylation and nutrient supply mechanism could collectively maintain higher levels of TCA cycle metabolites in cancer cells.

*7) The authors do not elucidate how the tumor cells uptake the nutrient loaded exosomes. The effects of heparin and CytoD should be shown also on the control condition, to ensure that things like proliferation inhibition are specific to the deprivation/restoration with exsomes. The use of macropinocytosis inhibitors (EIPA) should provide an answer to this question. If the delivery of endosomal contents in the cell requires degradation of exosomes by fusion with lysosome (to release their cargo) in cancer cells, then treatment with drugs that block this process such as CQ or BafilomycinA should impair the phenotypes observed with cancer cells are treated with CDEs.*

We thank the reviewers and editors for this suggestion. We have added the control conditions in Figure 6 and Figure 7 in prostate and pancreatic cancer cells, respectively. Please see the revised figures. Additionally, as suggested by the reviewers, we performed experiments with macropinocytosis inhibitor (EIPA) and lysosomal degradation inhibitor chloroquine (CQ) and have included them in Figure 6 and Figure 7—figure supplement 1. As seen in the figures, EIPA inhibits CDEs-induced increased cell proliferation, thus suggesting that CDEs uptake in cancer cells is also through macropinocytosis along with other endocytosis pathways. Further, Figure 6 and Figure 1 suggest that lysosomal degradation may not be the only pathway for release of exosomal content; other mechanisms may also be responsible for release of exosomal content.

Figure 6: Subsection “CDEs can supply amino acids to cancer cells in a manner similar to micropinocytosis”, last paragraph: “However, this rescue effect is reduced to varying extents by adding CytoD, heparin and lysosomal degradation inhibitor choloroquine (Figure 6). Similarly, addition of macropinocytosis inhibitor EIPA also counters the rescue of CDEs under deprivation (Figure 6). These data suggest that uptake of exosomes and release of their cargo is necessary to rescue cell proliferation under nutrient deprived conditions.”

Figure 7: Subsection "CDEs supply metabolites to pancreatic cancer cells via Kras-independent mechanism", first paragraph:

“Further, since both EIPA and CytoD could also inhibit rescue effect of proliferation in BxPC3 and MiaPaCa-2 cells (Figure 7—figure supplement 1), suggesting that endocytosis pathway dependent on macropinocytosis and caveolae mediated endocytosis are also associated with uptake of CDEs in pancreatic cancer cells. Additionally, CQ reduced rescue of proliferation by CDEs, thereby suggesting that release of exosomal content through lysosomes may play a role in some cancer cells. Nevertheless, these data suggest that CDEs internalization may happen through various modes of internalization.”

*8) One of the sections points at excluding KRAS from playing a role in PDAC uptake of CDEs. However, to fully exclude KRAS role, the analysis should be based on more than comparison between two cell lines. For example, if the phenotype (decreased OCR or increased proliferation upon treatment with CDEs) is maintained after genetic ablation of KRAS in MiaPaCa cells.*

Thank you for this suggestion. To exclude KRAS regulated effect on CDEs uptake in PDAC, we used doxycycline inducible *Kras**-***G12D cell line (iKras^-1^, (Ying et al., 2012)) and measured proliferation of the cells with and without CDEs. As seen in Figure 7—figure supplement 2, CDEs could rescue loss of proliferation with and without oncogenic Kras expression, thereby suggesting that internalization or uptake and supply of exosomes derived metabolites in cancer cells is Kras independent. We have added the results in Figure 7—figure supplement 2 and the description in the second paragraph of the subsection “CDEs supply metabolites to pancreatic cancer cells via Kras-independent mechanism“:

To further substantiate the role of KRAS, we used doxycycline inducible *Kras**-***G12D cell line (iKras^-1^, (Ying et al., 2012)) to test if enhancement of proliferation by CDEs indeed was KRAS independent. As shown in Figure 7—figure supplement 2, iKras^-1^ cells with or without doxycycline showed similar proliferation increases with CDEs, thereby suggesting that internalization or uptake and supply of exosome-derived metabolites in cancer cells is Kras independent.

9) The mitochondrial membrane potential is decreased in PC3 cells upon treatment with CDEs (Figure 2), did the authors verify if total mitochondrial content and/or shape is altered in cancer cells treated with CDEs? Also, did the authors verify if CDEs contain mitochondria or fractions of mitochondria? This could explain the relatively high levels of malate and other TCA cycle intermediates found inside (Figure 5).

As suggested, we measured total mitochondrial content through quantitative PCR by measuring expression level of mtDNA in cancer cells cultured with CDEs. Our data suggest that there is an insignificant change in total mitochondrial content in cancer cells with CDEs as seen below. Since the size of mitochondria is much bigger than exosomes, CDEs may not contain intact mitochondria. However, we did detect mitochondrial DNA in CDEs as seen in Figure 9. We have not measured mitochondrial fractions in CDEs and feel that it will be beyond the scope of current work.

Author response image 1.DNA extracted from whole CAFs (samples 1-4) or CAF-derived exosomes (samples 5-8) was analyzed for the presence of 5 mitochondrial genes, MT-ATP6, MT-ND5, MT-RNR1, MT-CYB and MT-CO3 by PCR.MT-ATP6: Mitochondrially Encoded ATP Synthase 6; MT-ND5: Mitochondrially Encoded NADH Dehydrogenase 5; MT-RNR1: Mitochondrially Encoded 12S RNA; MT-CYB: Mitochondrially Encoded Cytochrome B; MT-CO3: Mitochondrially Encoded Cytochrome C Oxidase III.**DOI:**
http://dx.doi.org/10.7554/eLife.10250.020

[Editors' note: further revisions were requested prior to acceptance, as described below.]

*The manuscript has been improved. However, there are significant remaining issues that we would like for you to provide a response, within 1-2 weeks, on how you would address them to determine whether a second revision is invited. While the effect of CAF-derived exosomes on OXPHOS (Figure 7) is impressive, the mechanisms by which this occurs remain undefined (issue 1, below), particularly with the issues surrounding the exosome metabolome (issue 2, below).*

We have appropriately addressed the issue 2 below. It was an inadvertent error that resulted in higher concentrations. Please see below our response.

1) The miRNA data is still very correlative and quite weak (based on literature showing that particular miRNAs can regulate expression of various oxphos enzymes).

We agree that the miRNA data is correlative and transfecting multiple miRNAs into the cells to show mechanism has not been possible due to cells dying during multiple transfections. In the first submission we had included the miRNA data in supplementary text and in the Discussion. However, reviewers found that data to be interesting and asked us to bring it into the main text. Hence, in our revisions, we included the data in the main body. We had mentioned in our response that there were technical difficulties due to limitations in genome engineering technologies to either insert multiple miRNAs in exosomes or transfect multiple miRNAs together without causing significant toxicity to the cells. The miRNA literature suggests that often miRNAs act in clusters, however there are inherent technological limitations in co-transfecting multiple miRNAs into our exosome-based cell assays as we found these to be toxic to the cells.

In order to elucidate the effect of miRNAs on the oxidative phosphorylation, we have included the preliminary data obtained by co-transfecting highly expressed miRs in exosomes in PC3 cells as Figure 2. In this figure, we measured oxygen consumption rate in PC3 cells co-transfected with miR22, let7a and miR125b and observe a significant reduction of OCR. In addition to these results, we have included the aforementioned caveats of co-transfecting miRNA in the Discussion section. We hope that the editors will agree that this data supports our hypothesis and conclusion.

The miRNAs and their targets that we have included are based on experimental miRNA abundance data from Nanostring assays followed by miRNA target prediction integrated with AGO-CLIP-SEQ data (Li et al., 2014). Therefore, the target genes we have identified are only those that have experimental evidence of interaction. We believe that our analytical approach and data is standard practice and with strong bioinformatic support lays the foundations for future experiments.

Inserted in the subsection “CDEs downregulate mitochondrial function of prostate cancer cells“: “The miRNAs and their targets that we have identified are based on experimental miRNA abundance data from Nanostring® assays followed by miRNA target prediction integrated with AGO-CLIP-SEQ data (Li et al., 2014). […] Although, the reduction of OCR is moderate, this is due to the technical limitation of co-transfection experiments, which can only allow using a small subset of the miRNAs that target OXPHOS in PC3.”

Inserted in the Discussion: “In Figure 2, we observe a reduction of OCR in PC3 cells co-transfected with miR-22, let7a and miR-25b, in line with our hypothesis of the inhibitory effect of miRNA from CDEs. […] In light of these results we believe further studies are needed to uncover the mechanism of ETC inhibition by CDEs-derived miRNAs that are out of scope for this study.”

Inserted in Figure 2 legend: “K. OCR of PC3 were measured after transfection of targeted miRNAs together into cells. (n=5). miRNAs were transfected into cells according to the manufacturer’s protocol (Lipofectamine® 2000 Transfection Reagent, Thermo fisher, cat. 11668-027). Cells were seeded in 6-well plate for 24h. Transfection was performed followed by incubation for 48h. Cells were then reseeded onto Seahorse plates for OCR measurements after the cells were attached.”

*2) The concentrations of metabolites in the exosomes continue to be unrealistic, e.g. 1 moles glutamine, 5M lactate moles, per g protein, which would translate to (assuming exosomes are 20% protein), 200M and 1000M, which is physically impossible. What is the relationship between exosome protein, exosome volume, and exosome number? Complete analytical support is necessary, including raw data, for any metabolites that the authors claim are present in exosomes at concentrations exceeding 20 mM.*

We thank the reviewers for pointing out this major issue in our data. We would like to apologize to the editors and reviewers for this confusion. Since the *eLife* required high quality figures during revisions, we used a newer version of illustrator on a computer that was missing several fonts. Hence, due to missing embedded fonts in the PDF files uploaded during revisions, several axis labels showing concentrations in µmol (micromoles) were converted automatically to mmol (milimoles). Due to this error, the data has been misinterpreted as being 1000 times higher than actual measurements. We would like to emphasize that this error only occurred in the revised submission and the original submission made in August 2015 had the correct units. The files on *eLife* server from August full submission can be checked for verifying our error. Further, we would be happy to provide the raw data and subsequent analyses of the data to definitively prove that this error occurred due to erroneous PDF files. Please note that we have made corrections in Figure 5, Figure 4, Figure 3, Figure 2 and Figure 1 on this regard.

We would like to refer the editors and reviewers to the table below, which shows the comparison of exo-metabolome concentrations with intracellular concentrations of metabolites obtained from previous studies. As seen in the table, our exo-metabolite concentrations are in similar ranges as reported previously for intracellular metabolites.

C: intracellular metabolite concentration, mM;

P: cells protein content, which is 0.2 g/mL (20% protein);

X: metabolite concentration normalized with protein amount, μmol/g protein

X=C ∙mmol(0.2g/mL)L=C ∙1000μmol(200g/L)L=5CμmolgFor example, in (Fan et al., 2013),Gluintra=37∙mmol(0.2g/mL)L=37∙1000μmol(200g/L)L=5×37μmolg=185μmolg

Amino acidMinimum concentration in Exo (μmol/g protein)Maximum concentration in Exo (μmol/g protein)Intracellular concentration obtained from previous studies (assuming 20% protein) (μmol/g protein)ReferencesGlutamate30.8294585.3429185(Fan et al., 2013)114.75(Baydoun et al., 1990)Phe3.5357223.73383.8(Baydoun et al., 1990)3.9(Hansen and Emborg, 1994)Alanine39.4691191.361214.75(Baydoun et al., 1990)Glycine60.8029168.640731.35(Baydoun et al., 1990)41.8(Hansen and Emborg, 1994)Tryptophan44.596451.1827.8(Baydoun et al., 1990)Asparagine49.7664133.011793.35(Hansen and Emborg, 1994)Leu10.3048174.23126.2(Baydoun et al., 1990)Valine90.0222196.7996.15(Baydoun et al., 1990)Lys24.451978.55655.4(Baydoun et al., 1990)Gln462.80791693.0858100(Souba, 1993)

Importantly, our exosome metabolome data is accurate when measured in either micromoles or micrograms or micromoles/g protein. To avoid issues related to normalization, we propose to estimate the concentration of metabolites with respect to exosome particle number rather than exosomal protein. This will avoid the bias created by the apparently large concentrations of metabolites within exosomes. It is also in line with the preference of normalizing metabolite concentrations with cell number in the field of cell metabolism. To the best of our knowledge, most of the metabolomics and metabolic flux literature use cell number as the normalization factor.

As reported previously by us, protein content in exosomes is 4.9μg/ 10^[9]^ particles ((El-Andaloussi et al., 2012), Nature Protocol estimates this number to be between 2-8μg/ 10^[9]^ particles).

Furthermore, we may expect higher concentrations in vesicles. It is known that during receptor mediated endocytosis there is a significant increase in concentration. Another example of this phenomenon is the glutamate concentration inside synaptic vesicles, which can be as high as 100 mM (Danbolt, 2001). A previous study has stated (Mavelli and Stano, 2015) that during the process of micro-vesicles formation it could be possible for solutes, in this case metabolites, to become concentrated within vesicles as compared to within cells. In the aforementioned study, the authors observed this phenomenon in liposomes that are similar in size to exosomes. Hence, it is very likely that during vesicle formation process, concentrations of metabolites, proteins, etc. become higher than that in the cells. To the best of our knowledge, not much is known about concentration estimation inside exosomes. Collectively our data shows the strong effect of CDEs in the form of (a) rescue of cell viability under nutrient deprivation; (b) metabolic reprogramming of OXPHOS and reductive carboxylation; (c) cargo that supplies intact metabolites to the TCA cycle.

*In addition to the two major issues above, there are other items that need substantive addressing. 1) Figure 6 seems confusing. Since lysine is an essential amino acid, why are they seeing M+2 form? Why do lactate, pyruvate, and alanine, which all share the same carbon skeleton, all have different labeling patterns? How do they get to the conclusion that ~ 30% of KG is derived from exosome metabolites? It does not seem proper to normalize the KG labeling in cells to that in exosomes, when most KG in cells comes from glutamine or glutamate. It is also confusing how labeling of KG in cells would exceed glutamine and glutamate. I do not want to cause the authors excessive suffering on this point. To this end, they may be better served reporting less isotope labeling data, instead of reporting dubious data. The leucine data, for example, makes the point that the exosome contribution is modest while looking more straightforward than some other data. Similarly, it seems important to see results for glutamine. Perhaps a few well-chosen examples would suffice. Also, perhaps some funny labeling patterns can be cleaned up by more careful examination of the raw data to eliminate interfering peaks.*

We thank the reviewers for their suggestion in improving the clarity of data shown in Figure 6. We agree with the reviewers that indeed there are a few puzzling trends seen in the mass isotopologue distributions of lactate, pyruvate, alanine and lysine. We agree with the reviewer’s concern about the dissimilarity in the labeling pattern of pyruvate, alanine and lactate. This may be due to combined effect of compartmentalization of pyruvate pools, complex carbon transitions in TCA cycle and malic enzyme pathways and multiple sources of pyruvate leading to distinct labeling patterns in alanine and pyruvate. Since this experiment was performed as a proof-of-concept for showing that CDEs contribute TCA cycle metabolites to cancer cells, we now show the mass distributions of the heaviest isotopologues. The data now highlights M6 Glucose derived M3 lactate, M3 pyruvate and M3 alanine that can only be directly sourced from exosomes. Similarly, we only show M5 Glutamine, M5 Glutamate, M6 lysine, M6 leucine indicating the fraction of these metabolites derived from exosomes. We have modified the text for Figure 6 to reflect the changes in the figure (subsection “CDEs can supply amino acids to cancer cells in a manner similar to micropinocytosis”). If the reviewers are in agreement, to enhance clarity for the reader we propose to report only the ^[12]^C and ^[13]^C fractional enrichment for these metabolites as has been done in previous studies showing metabolite-tracing isotopic data.

In the case of citrate, α-ketoglutarate and malate, we show complete mass distributions as they are important TCA cycle metabolites, which are enriched by exosome-derived metabolites metabolized inside cancer cells. Due to branching and high turnover rates in the TCA cycle it is important that we show their complete mass distributions, which also support our hypothesis that exosome-derived metabolites are incorporated into the TCA cycle. Furthermore, we agree that the α-KG contribution is 30% and much higher than that of glutamine and glutamate, it is due to the low 13C enrichment of α-KG in exosome, which results in overestimating its contribution. We had reported these values based on the suggestion made in the previous review to quantify the contribution of exosome metabolites to cancer cells. It is important to note here that due to complexity of TCA cycle pathways it becomes extremely difficult to resolve contribution of metabolites that comes directly from exosomes and that from metabolites metabolized after uptake of exosomes by cancer cells. Hence, as suggested we have removed the figure (Figure 6—figure supplement 1, Panel B) that reported the estimated contributions of TCA metabolites by normalizing mean enrichment with exosomes enrichment.

*2) The authors provide some more rationale for choosing the concentration of exosomes used in their studies. However, there is still uncertainty as to how this represents the physiological state. The authors should at least write a sentence or 2 in the Discussion talking about this limitation. The limitation of this analysis (protracted purification protocol) should be mentioned in the Discussion.*

As per the reviewers’ suggestion, we have added a paragraph addressing this issue. We would like to stress that the physiological value of exosome concentrations should be considered from a perspective of a coculture or an organotypic model of tumors, which can mimic the in vivo ratio of stromal cells to cancer cells. The effect of tumor microenvironment cells on cancer cells can only be dissected accurately using a tumor-mimicking design. Apart from these considerations, previous studies have also reported similar concentrations of exosomes in tumors (Baranyai et al., 2015).

We have used 200 μg/ml for the different types of experiments conducted herein (proliferation assays, metabolic assays and tracer experiments). The working concentration was chosen to maintain a physiological ratio of CAFs to cancer cells which is reported to be between 1 and10 (Brauer et al., 2013; Delinassios, 1987; Hu et al., 2015). A concentration of 200 μg/ml corresponded to ratio between 1 and 5 CAFs per cancer cell and hence, is within the range reported in the literature. However, for different CAFs and cancer cells system this ratio should be individually estimated based on the functional effect of stromal cells on cancer cells. It is to be noted that due to protracted purification steps during exosome isolation, degradation of metabolites can occur and hence, replenishment of exosomes for functional studies was followed and recommended. Additionally, to minimize degradation of metabolites in exosomes, we used fresh exosomes for all the experiments and avoided any freeze-thaw cycle in exosomes that were introduced to cancer cells cultures.

*3) The correction of the lactate levels to 1/10th what was originally reported does not seem "minor". Can the authors please explain this further?*

During post-processing of all of our metabolites data whether it was obtained from UPLC, GC-MS or ELISA, we had initially used either mL or L (liters) for normalization. However, lactate measurements were obtained from colorometric assay which uses Trinity Bio's standard. This was the only standard that was in mg/dl instead of mg/l due to which an inadvertent error occurred. Hence, during normalization, lactate concentrations were ten times higher.Further, we would be happy to provide the raw data and subsequent analyses of the data.

*4) It is still not clear why M+2 is the dominant citrate isotopomer in Figure 6—figure supplement 1. This continues to suggest that the majority of this citrate is not coming directly from the exosome labeled citrate but rather from Acetyl-CoA (or precursors) delivered by exosomes.*

The reviewers make an appropriate observation that M2 citrate is the dominant isotopomer suggesting the contribution of Acetyl-CoA. We would like to point out that this data is meant to indicate the contribution of labeled metabolites (Glucose, Glutamine, Lysine, Phenylalanine, Pyruvate, and Leucine) derived from exosomes and their subsequent incorporation into central carbon metabolism. The TCA cycle metabolites citrate, α-ketoglutarate and malate are enriched by exosome-derived metabolites that are metabolized inside cancer cells. We would like to clarify that the contribution of citrate is not all directly from exosomes, rather most of the citrate is from other metabolites supplied by exosomes such as labeled pyruvate/acetyl-CoA (from glucose) and labeled malate (from glutamine and other precursors).

*5) In the bar graphs looking at drug effects, the authors should not limit tests of statistical significance to the hypothesized effects (in the exosome rescue conditions), but also conduct the same tests in the non-rescue conditions, as it seems that the drugs sometimes have an effect also in those conditions. If those tests were all non-significant, there is no need to mark them on the figure, but that should be clearly reported.*

We thank the reviewers for this suggestion and have addressed this issue by inserting the statistics in the control conditions. We have revised Figure 6, Figure 7 and Figure 7—figure supplement 1.

6) The authors should be aware that oxoproline is most commonly found due to degradation of glutamate during sample processing. They should not make main text statements relating this to the glutathione pathway without further evidence.

We have deleted the associated text in the revised manuscript.